# Spatio-Temporal Evolution of Ecological Resilience in Ecologically Fragile Areas and Its Influencing Factors: A Case Study of the Wuling Mountains Area, China

Jilin Wu [1,2], Manhong Yang [1], Jinyou Zuo [1], Ningling Yin [1], Yimin Yang [1], Wenhai Xie [1,2] and Shuiliang Liu [3,*]

1    School of Civil Engineering and Architecture, Jishou University, Zhangjiajie 427000, China; wjlin1978@jsu.edu.cn (J.W.); 2021700527@stu.jsu.edu.cn (M.Y.); 2020700503@stu.jsu.edu.cn (J.Z.); 2021700529@stu.jsu.edu.cn (N.Y.); 20155195038@stu.jsu.edu.cn (Y.Y.); xiewhai@jsu.edu.cn (W.X.)
2    Rural Planning and Development Research Center of Wuling Mountain Area, Zhangjiajie 427000, China
3    College of Tourism, Jishou University, Zhangjiajie 427000, China
*    Correspondence: lsliang7412@jsu.edu.cn; Tel.: +86-13637449949

**Abstract:** The ecological environment of the Wuling Mountains region has been impacted by climate change and economic development, necessitating immediate reinforcement of ecological protection and restoration measures. The study utilized the normalized vegetation index (NDVI) as a proxy for ecological resilience. NDVI data from 2000 to 2020 were employed to compute the ecological resilience index of the Wuling Mountains area and to examine its spatial and temporal evolution as well as the factors influencing it. The findings indicate that: (1) The ecological resilience index increased in the Wuling Mountains area and Guizhou, Chongqing, and Hunan sub-areas but decreased in the Hubei sub-area. (2) The ecological resilience varies significantly in the Wuling Mountains area and the Guizhou, Hubei, and Hunan sub-regions, whereas it varies less in the Chongqing sub-region. (3) The primary elements influencing the ecological resilience capability of the Wuling Mountains area and its four sub-areas are climate conditions and socio-economic factors, respectively. The study can offer a scientific foundation for ecological conservation and restoration efforts in the Wuling Mountains area, as well as serve as a benchmark for measuring ecological resilience in other environmentally vulnerable regions.

**Keywords:** ecological resilience; spatio-temporal evolution; influencing factors; Wuling Mountains area

## 1. Introduction

Ecologically fragile places are transitional zones between distinct landscapes or ecosystems [1]. These regions often face ecological issues caused by low ecosystem resiliency, stability, notable edge effects, and limited self-recovery capability [2–4]. Additionally, ecologically vulnerable areas are often ecological barriers that maintain the stability and diversity of surrounding ecosystems [5]. Therefore, protecting the ecological environment is crucial in these areas because they are focal points for ecological restoration efforts [6]. Resilience, a fundamental characteristic of ecosystems, is frequently employed in ecological restoration initiatives to evaluate an ecosystem's ability to withstand and bounce back from disruption [7]. Assessing the ecological resilience of vulnerable places is crucial for enhancing regional resilience and sustainability.

In 1973, Holling proposed the resilience of ecosystems, defining ecological resilience as a measure of the adaptive capacity of ecosystems [8]. The term describes a system's capacity to sustain interactions among populations or state variables within ecosystems and to preserve system performance in the face of natural and human disruptions. Ecological resilience has been widely investigated in sustainable development research, with scholars assessing regional ecological resilience at different scales. Examples consist of evaluations on ecological resilience in huge marine ecosystems [9], ecological network resilience in

urban agglomerations [10], research on the ecological resilience of resource-based cities [11], and assessments on rural ecological resilience [12]. Scholars have studied the effects of climate change and economic development on ecological resilience. This involves studying the connection between urbanization and ecological resilience [13], the impact of urbanization on ecological resilience [14], and the spatial and temporal changes in environmental quality and ecological resilience patterns [15]. The differences and convergences in regional ecological resilience are also a research focus [16]. Despite abundant research on ecological resilience, few studies have investigated the ecological resilience of ecologically fragile areas. Therefore, there is an urgent need for theoretical and practical research on this topic.

Methods for measuring resilience include establishing thresholds [17], scenario analysis [18], state space analysis [17], experiments [19], and resilience substitution [20,21]. Folke considered resilience and vulnerability as two sides of the same coin and viewed vulnerability as the opposite of resilience [22]. Ecological resilience is negatively correlated with ecological sensitivity and vulnerability [7]. Ecological vulnerability has a negative correlation with sensitivity and a positive correlation with adaptation [20,21]. Due to the intrinsic properties of resilience, direct measurement of resilience is very difficult [23], the resilience substitution was derived from this context. The principle of the resilience substitution method is to identify attributes related to resilience in the system that are easy to measure, select substitute factors for resilience from them, and indirectly measure resilience [24].

Some scholars have selected land use data instead of ecological resilience to construct ecological resilience evaluation models [25]. Ecosystem resilience has been assessed by analyzing the variation, stability, resistance, and maximum value of Net Primary Productivity (NPP) [26]. Alterations in the spatial and temporal arrangement of vegetation impact the composition and operation of landscapes, influencing ecological mechanisms [27]. Vegetation reflects the regional climate, soil, and hydrological conditions; thus, it is a crucial indicator of regional ecological quality [28]. The Normalized Difference Vegetation Index (NDVI) is utilized as an ecological sensitivity indicator [10]. The NDVI is highly sensitive to vegetation cover and dynamics and can capture subtle changes in vegetation conditions over time, reflecting changes in the regional ecological environment. NDVI has been applied to identify ecologically sensitive areas [29], to track ecosystem dynamics and environmental management [30], or as one of the indicators to evaluate ecological resilience [31,32]. Previous research often assessed ecological resilience by creating a whole assessment index system. However, the exploration of utilizing NDVI as a sole indicator for measuring ecological resilience still need more development.

Scholars have extensively researched the aspects that influence ecological resilience. Variations in natural and socio-economic conditions between areas lead to different factors affecting ecological resilience on a regional level. Economic factors influence urban resilience [33]. Temperature and precipitation have significant impacts on ecological resilience, and human activities have increasingly affected ecological resilience [25]. Altitude, slope, and GDP are positively correlated with ecological resilience, whereas precipitation, temperature, population density, and construction land are negatively correlated with ecological resilience [21]. Techniques for examining the factors that affect ecological resilience include grey relational analysis [33], geographically weighted regression models [34], multiple linear regression models [21], and geographic detectors [25]. The advantage of geographic detectors is that the data do not have to be normally distributed, and parameter setting is less complex. Additionally, multicollinearity among the input factors is not a problem, and increasing or decreasing the factor values does not affect the other factors' results [35–37].

The Wuling Mountains area is located in the ecologically sensitive southern red soil hilly region and the southwestern karst mountain rocky desertification area in China. This region is susceptible to natural disasters and exhibits a high level of ecological sensitivity [38]. It is situated in the confluence of Guizhou Province, Chongqing City, Hubei Province, and Hunan Province. The variations in the ecological environment are intricately linked to the natural and socio-economic variables of the four provinces (municipalities).

Thus, examining the biological and environmental issues in the Wuling Mountains area necessitates combining both the area and sub-area levels. However, previous studies have concentrated on the effects of urbanization and tourism growth on the ecosystems of the Wuling Mountains region [39,40], whereas few studies have evaluated the ecological resilience of this area.

The ecological environment in the Wuling Mountains area is delicate, with limited self-healing capabilities in the ecosystem. This study employs ecological resilience to assess the ecological environment of the Wuling Mountains area in order to aid in ecological protection and promote high-quality development, focusing on the following three issues: Firstly, how to precisely evaluate the ecological resilience of the Wuling Mountains area; Secondly, what is the variability in the ecological resilience of the Wuling Mountains area? Thirdly, what are the primary variables influencing the ecological resilience of the Wuling Mountains area? NDVI was chosen as a proxy for ecological resilience because of the mentioned issues. The ecological resilience index was computed using the NDVI data, and the spatial and temporal divergence features of ecological resilience in areas and sub-areas were examined using a geodetector to explore the main influences affecting ecological resilience in patches and sub-patches. Finally, recommendations for enhancing ecological resilience are provided based on the findings of the influencing factors study. This study offers a quantitative assessment approach for ecological resilience in the Wuling Mountains area. It gives theoretical and practical insights for evaluating ecological resilience in similar ecologically vulnerable regions. The research findings offer a scientific foundation for ecological protection and restoration in the Wuling Mountains area and are crucial for advancing sustainable development in environmentally vulnerable regions.

## 2. Materials and Methods

### 2.1. Study Area

The Wuling Mountains area is situated in the intersection of Hunan, Hubei, Guizhou, and Chongqing, spanning 71 counties (cities, districts) and covering a total area of 171,800 km$^2$ (Figure 1). The landscape consists mainly of mountains and hills, with steep slopes, an average elevation of about 1000 m, and a complex geographic environment. The area experiences a subtropical monsoon climate, with an average annual temperature ranging from 12 to 17 °C and an annual precipitation between 1100 and 1600 mm. Some areas are prone to soil erosion and rocky desertification, resulting in a fragile ecological environment [41]. Additionally, the Wuling Mountains area is an important ecological security barrier in the Yangtze River basin and is crucial to national ecological security.

However, rapid economic development and accelerated urbanization have negatively impacted the ecological environment, threatening regional ecological security [42,43]. From 2000 to 2020, the GDP per square kilometer in the Wuling Mountains area increased from 59.79 thousand yuan to 595.92 thousand yuan, and the proportion of construction land has risen from 15.35% to 34.41%. The Wuling Mountains area encounters difficulties in harmonizing economic development with environmental protection, akin to ecologically vulnerable regions worldwide. Studying ecological resilience in the Wuling Mountains area can provide a model for resolving ecological fragility in other locations.

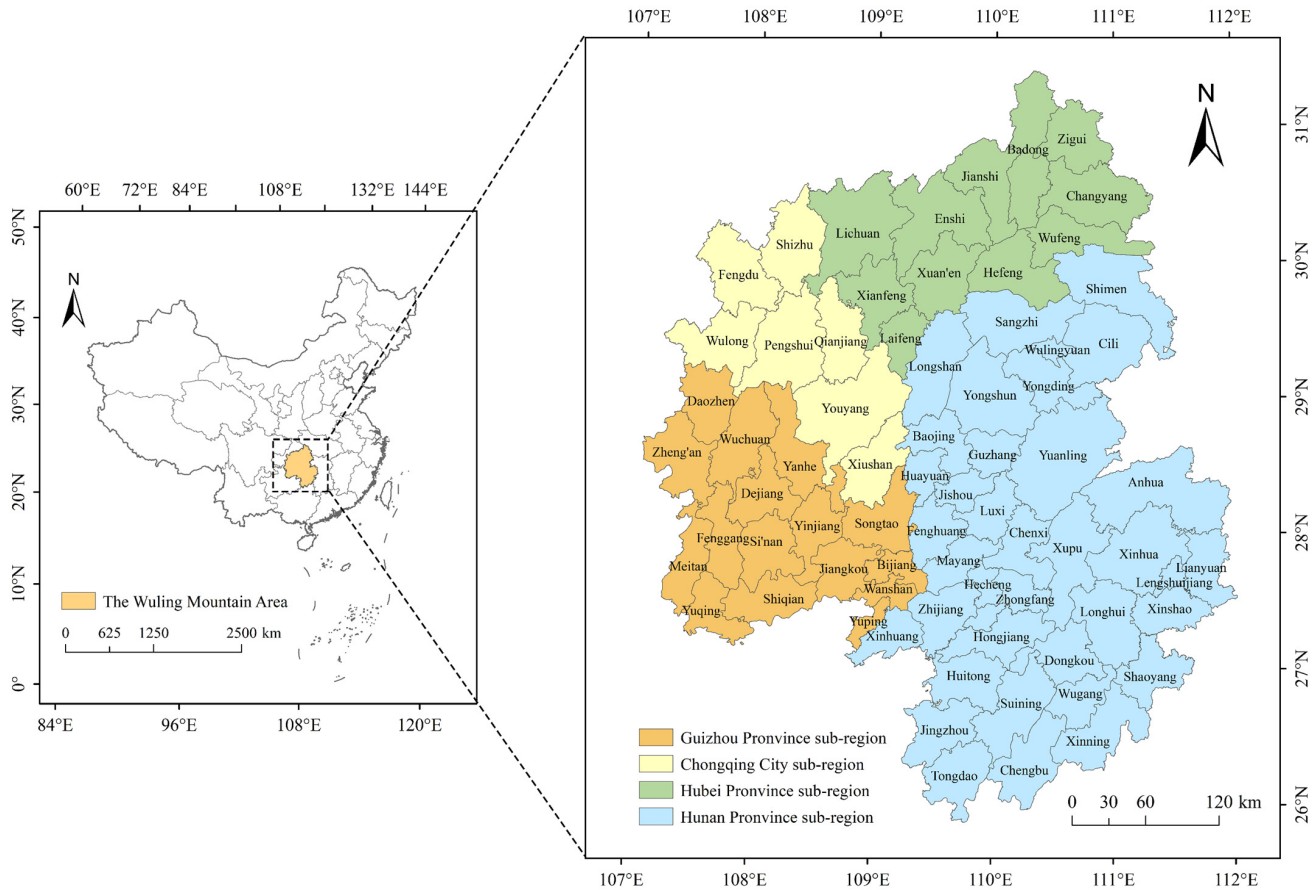

**Figure 1.** Geographical location and administrative subdivisions of the Wuling Mountains region. The map was obtained from the Standard Map Service website of the Ministry of Natural Resources of China, with the map number GS (2020) 4619. The map boundaries remained unchanged.

### 2.2. Selection of Influencing Factors

Natural factors are the basis of the spatial and temporal distribution of ecological resilience. Temperature and precipitation are primary factors influencing the spatial distribution of vegetation cover, which significantly influences ecological resilience [44]. Topography influences the spread of different land use categories, impacting the ability of ecosystems to recover and adapt [45]. Ecological resilience is affected by natural elements such as climate change and terrain [46]. Therefore, annual precipitation, average annual temperature, slope, and elevation were selected as geographical factors (Table 1). Economic factors also affect ecological resilience. The population density affects industrial development and resource recycling, but a high population density can cause ecosystem destruction and pollution. The regional average population density can effectively reflect the degree of regional population agglomeration [16,47]. The average GDP of a region reflects its economic development level and influences ecological resilience [16,46]. The proportion of built-up land area can reflect the extent to which people develop and utilize the land [48]. Carbon dioxide emissions reflect the regional green economy and sustainable development level [21,49]. Therefore, population density, GDP, proportion of construction land, and carbon emissions were selected as socio-economic factors (Table 1).

**Table 1.** Indicators of Ecological Resilience in the Wuling Mountains Area.

| Type | Influencing Factors | Indicators | Unit |
|---|---|---|---|
| Physical and geographic factors | X1 | Average annual temperature | °C |
| | X2 | Annual precipitation | mm |
| | X3 | Slope | ° |
| | X4 | Elevation | m |
| Social and economic factors | X5 | Population density | Person/km$^2$ |
| | X6 | GDP | 10,000 yuan/km$^2$ |
| | X7 | Proportion of construction land | % |
| | X8 | Carbon emissions | 10,000 tons |

*2.3. Data Sources and Preprocessing*

2.3.1. Raster Data

The NDVI, annual mean temperature, DEM, annual precipitation, GDP, population density, and land cover (LUCC) data are raster datasets sourced from the Resource and Environmental Science and Data Centre (RESDC) of the Chinese Academy of Sciences. Due to data limitations, 2019 population density data and GDP data were used instead of 2020 population density data and GDP data [21]. All these raster data need to be cropped, projected, and resampled using ArcGIS10.6 software, respectively, cropping them out of the Wuling Mountains area, projecting them to the same projection and coordinate system (UTM map projection and GCS_WGS_1984 coordinate system), and uniformly sampling them to a resolution of 1 km × 1 km. Table 2 displays the data sources.

**Table 2.** Data Sources for the Study Area.

| Data Name | Time (Year) | Resolution | Data Source |
|---|---|---|---|
| NDVI | 2000–2020 | 30 m | RESDC |
| DEM | \ | 250 m | RESDC |
| Average Annual Temperature, Precipitation | 2000, 2005, 2010, 2015, 2020 | 1 km | RESDC |
| LUCC | 2000, 2005, 2010, 2015, 2020 | 250 m | RESDC |
| GDP, Population Density | 2000, 2005, 2010, 2015, 2019 | 1 km | RESDC |
| Carbon Emissions | 2000–2017 | \ | CEADs |

The NDVI data require the use of ArcGIS10.6 software to calculate their raw values and remove outliers (see Figure 2). ArcGIS10.6 software was utilized to compute the Normalized Difference Vegetation Index (NDVI) for 71 counties in the Wuling Mountains area.

Land use data need to be categorized into arable land, forest land, grassland, watershed, building land, and unused land using ArcGIS10.6 software. Then, ArcGIS 10.6 software was used to calculate the construction land area share of 71 counties in the Wuling Mountains area.

The slope and elevation statistics were derived from calculations based on digital elevation model (DEM) data.

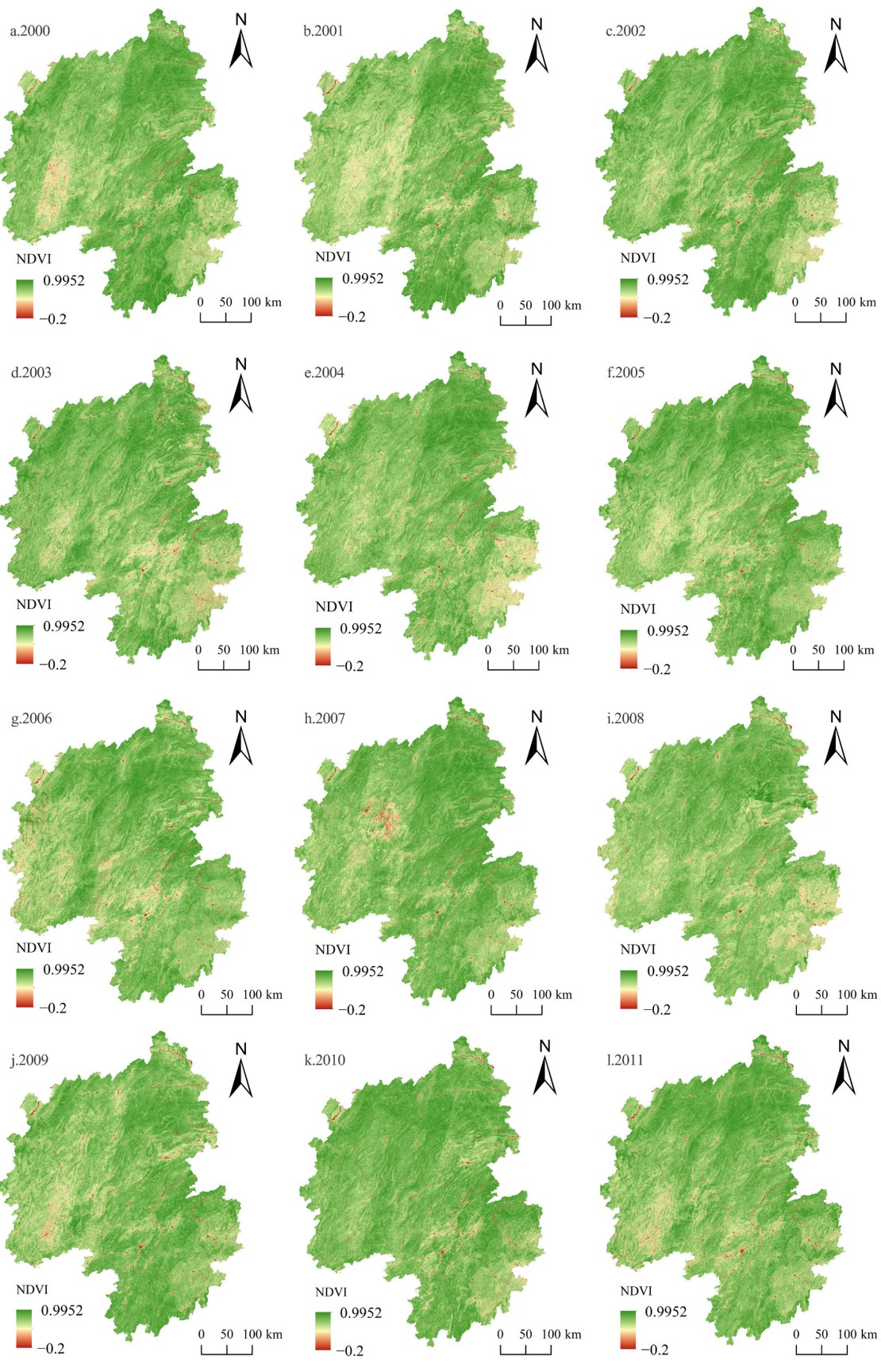

**Figure 2.** *Cont.*

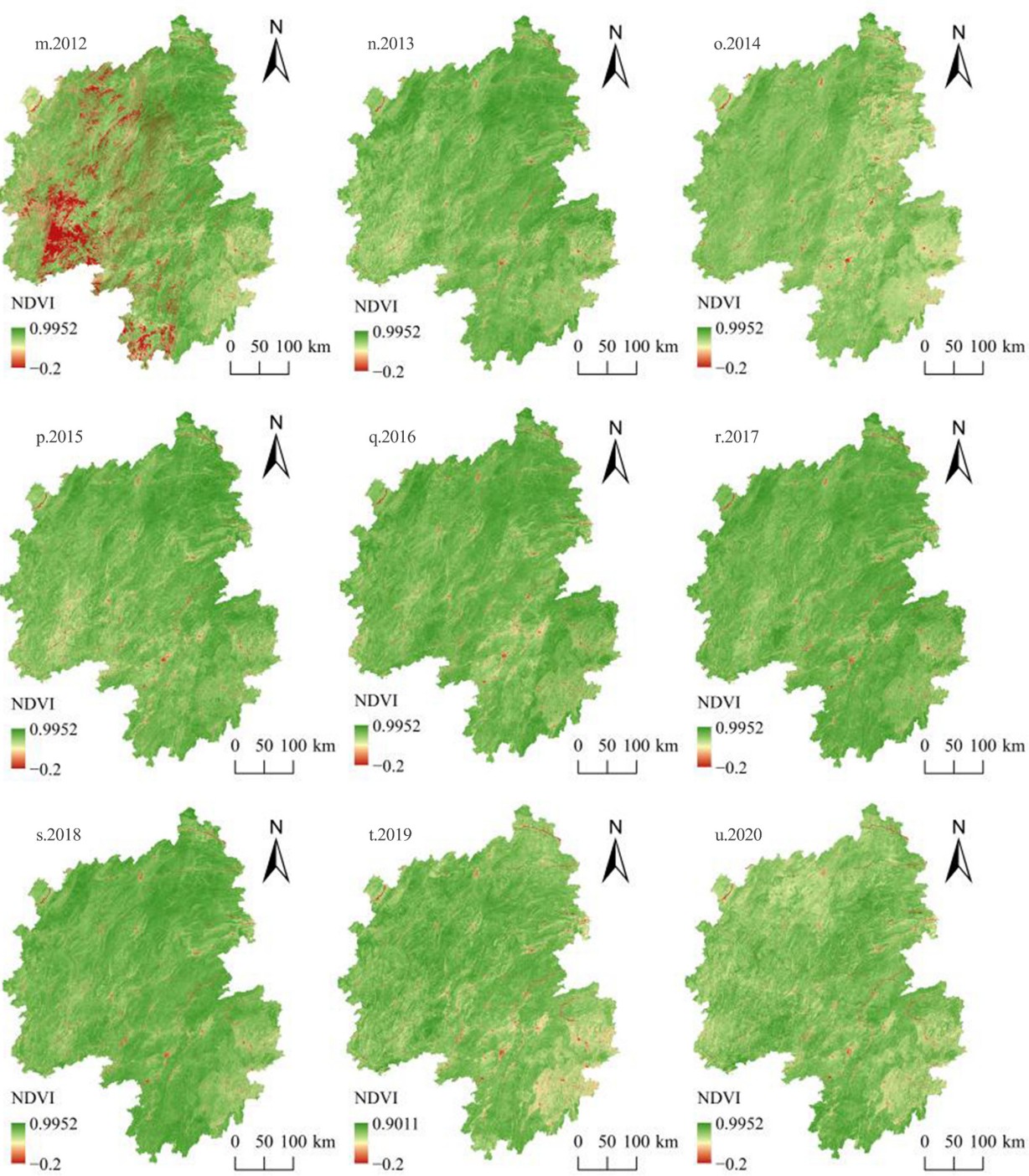

**Figure 2.** Distribution of NDVI in the Wuling Mountains Area, 2000–2020.

### 2.3.2. Other Data

Carbon emissions data were obtained from the China Carbon Accounting Database (CEADs). Since the county carbon emissions data are now updated only to 2017, this study uses the $CO_2$ emissions data of counties from 2000 to 2017 to obtain the carbon emissions data of 71 counties in the Wuling Mountains area in 2020 through the ARMA prediction model.

*2.4. Methods*

2.4.1. Estimation of Ecological Resilience

NDVI is a crucial indication of the ecological quality of a region. Studying the dynamic changes in NDVI can offer insights into how well regional ecosystems can maintain a dynamic equilibrium when faced with external disruptions. Therefore, NDVI was selected as a proxy for ecological resilience. NDVI data were utilized in this research to compute the sensitivity index, adaptability index, and ecological resilience index.

Sensitivity Index: It represents the system's response capacity to disturbances under normal conditions, specifically the response to climate change disturbances [50]. The system's sensitivity is indicated by the interannual fluctuation of the NDVI between 2000 and 2020, showing how much the NDVI deviates from its average value. The calculation formula is as follows:

$$S = \frac{\sum_{i=1}^{n} |p_i - p|}{p} \tag{1}$$

Equation (1) defines S as the sensitivity index, i as year i (n = 21), pi as the NDVI value in year i, and p as the mean NDVI value.

Adaptability Index: It represents the system's ability to maintain and recover its structure and function after a disturbance [51]. Ecosystem adaptability in a period can be measured by the deviation of the ecosystem's trend from the equilibrium. If the trend decreases or remains unchanged, the system tends to lean toward relative stability. Conversely, an unstable system shows adaptive changes, which may indicate increased fragility [22,52]. In this study, the adaptability of the ecosystem was represented by the slope of the linear trend line fitted to the interannual variation of the NDVI from 2000 to 2020 as follows:

$$y = Ax + B \tag{2}$$

$$A = \frac{n \sum xy - (\sum x)(\sum y)}{n \sum x^2 - (\sum x)^2} \tag{3}$$

In Equations (2) and (3): x signifies the years 2000, . . ., and 2020, y denotes the annual rate of change of NDVI, A is the adaptive index, and B denotes the intercept.

The ecosystem's resilience is determined by its traits, magnitude, and pace of change, which are influenced by its sensitivity and adaptability [53]. The adaptability, sensitivity, and resilience levels of each region vary, and the computed S and A values may not be in the same dimension as per Equations (1) and (3). It is necessary to standardize the sensitivity and adaptability indices before estimating ecological resilience [54]. The ecological resilience formula is as follows:

$$R = A - S \tag{4}$$

R represents the ecological resilience index, A represents the standardized adaptation index, and S represents the standardized sensitivity index.

2.4.2. Geodetector

The Geodetector is a statistical model used to analyze the impact of various causes by measuring regional heterogeneity. It presumes that two associated components have similar spatial distributions. The study used the Factor Detector to examine the influencing elements. Examining the impact of particular factors on ecological resilience utilizing the q-value [55]:

$$q = 1 - \frac{\sum_{h=1}^{L} N_h \sigma_h^2}{N \sigma^2} \tag{5}$$

In Equation (5): h denotes the classification of variable Y or factor X; Nh and N are the number of units in layer h and the whole area, respectively; $\sigma_h^2$ and $\sigma^2$ are the variance of Y values in layer h and the whole area, respectively. q value has a range of [0, 1], and a larger value of q indicates a stronger explanatory power for ecological resilience.

The independent variables must be categorical in the Geodetector. Therefore, numerical data must be discretized. We used natural breakpoints to discretize the data. The study area was sampled using ArcGIS10.6 software with a sampling interval of 3 km, resulting in 27,464 randomly generated sampling points. Ecological resilience was analyzed in connection to several parameters by analyzing their spatial correlations.

## 3. Results

### 3.1. Temporal Evolution Characteristics of Ecological Resilience in the Wuling Mountains Area

The ecological resilience index in the Wuling Mountains area showed an initial gain followed by a fall from 2000 to 2020, as shown in Table 3. Its values were 0.171, 0.209, 0.356, and 0.235 in 2000–2005, 2005–2010, 2010–2015, and 2015–2020, respectively. The coefficient of variation had a range of [0.245, 0.321], demonstrating an initial rise followed by a decline, indicating variability in the ecological resilience index and a decrease in dispersion. The global Moran's I values for the ecological resilience index in the four periods were 0.512, 0.461, 0.481, and 0.660, respectively. All results were significant, showing a notable positive spatial autocorrelation trend and an escalation in the spatial clustering degree of the ecological resilience index in the Wuling Mountains region.

**Table 3.** Changes in the Ecological Resilience Index and Moran's I Values in the Wuling Mountains Area from 2000 to 2020.

| Year | Ecological Resilience Index | | Spatial Autocorrelation Index | | |
|---|---|---|---|---|---|
| | Mean | Standard Deviation | Moran's I | Z Score | *p* Value |
| 2000–2005 | 0.171 | 0.297 | 0.512 | 5.792 | 0.000 |
| 2005–2010 | 0.209 | 0.314 | 0.461 | 5.238 | 0.000 |
| 2010–2015 | 0.356 | 0.321 | 0.481 | 5.643 | 0.000 |
| 2015–2020 | 0.235 | 0.245 | 0.660 | 7.478 | 0.000 |

Figure 3 displays the temporal evolution characteristics of ecological resilience in the four sub-regions. The ecological resilience index in the Guizhou Province sub-region increased from 2000 to 2020. Guizhou Province has carried out ecological restoration of open-pit mines and achieved remarkable results. The ecological resilience index increased from 0.006 in 2000–2005 to 0.307 in 2015–2020, a 50-fold increase. The ecological resilience index increased over time in the Chongqing City sub-region. Chongqing ensured soil and water conservation through afforestation and developed a green and low-carbon economy to reduce industrial pollution, resulting in improved environmental quality. The ecological resilience index increased twofold from 0.108 in 2000–2005 to 0.349 in 2015–2020. The ecological resilience index in the Hubei Province sub-region was relatively high but gradually declined. Because of the swift economic growth, the self-regulation of the ecosystem in Hubei Province has been affected, resulting in a decline in ecological resilience. The ecological resilience index decreased from 0.403 in 2000–2005 to 0.194 in 2015–2020, a decrease of 51.86%. The ecological resilience index in the Hunan Province sub-region generally increased. Hunan has a strong focus on environmental protection, and its regulations, systems, and policies are relatively sound. The ecological resilience index rose from 0.184 in 2000–2005 to 0.195 in 2015–2020, representing a 5.98% rise.

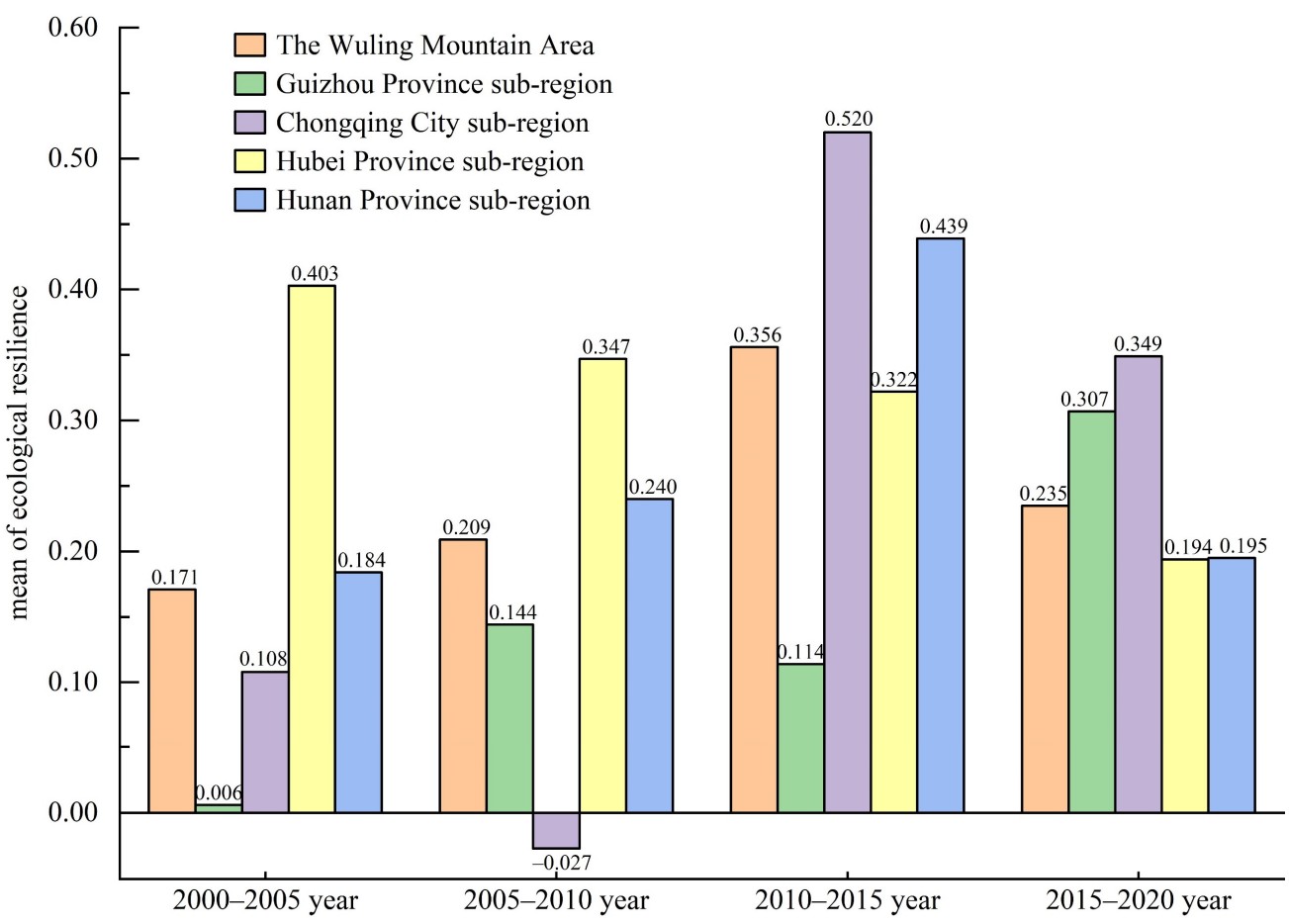

**Figure 3.** Temporal Evolution Characteristics of Ecological Resilience in the Wuling Mountains Area and Four Sub-Regions.

### 3.2. Spatial Evolution Characteristics of Ecological Resilience in the Wuling Mountains Area

The ecological resilience index was categorized into five categories using natural breakpoints (Figure 4). Figure 5 illustrates the progression of the ecological resilience index levels. Between 2000 and 2020, the ecological resilience of the Wuling Mountains area was mainly at medium and high levels. The spatial distribution of ecological resilience moved dramatically. The number of counties with low and high ecological resilience levels decreased. The number of counties with low–medium ecological resilience first decreased and then increased. The count of counties with medium and medium–high ecological resilience increased. Between 2000 and 2005, areas with medium–high and high ecological resilience in the Wuling Mountains area were primarily located in the Hubei sub-area and the northern part of the Hunan sub-area. Conversely, areas with low and low–medium ecological resilience were mainly found in the central part of the Guizhou sub-area, as well as in the southern parts of the Chongqing and Hunan sub-areas. The number of counties with medium ecological resilience was the largest; the ecological resilience of these counties is mainly transformed into low ecological resilience.

Between 2005 and 2010, places with medium–high and high ecological resilience in the Wuling Mountains region were primarily located in the Hubei sub-area, central Guizhou sub-area, and eastern, southern, and northern parts of the Hunan sub-area. Areas with low and low–medium ecological resilience were primarily found in the Chongqing sub-region, the western portion of the Hunan sub-region, and the eastern and western parts of the Guizhou sub-region. The number of counties with medium ecological resilience was the largest; the ecological resilience of these counties is mainly transformed into medium–high ecological resilience.

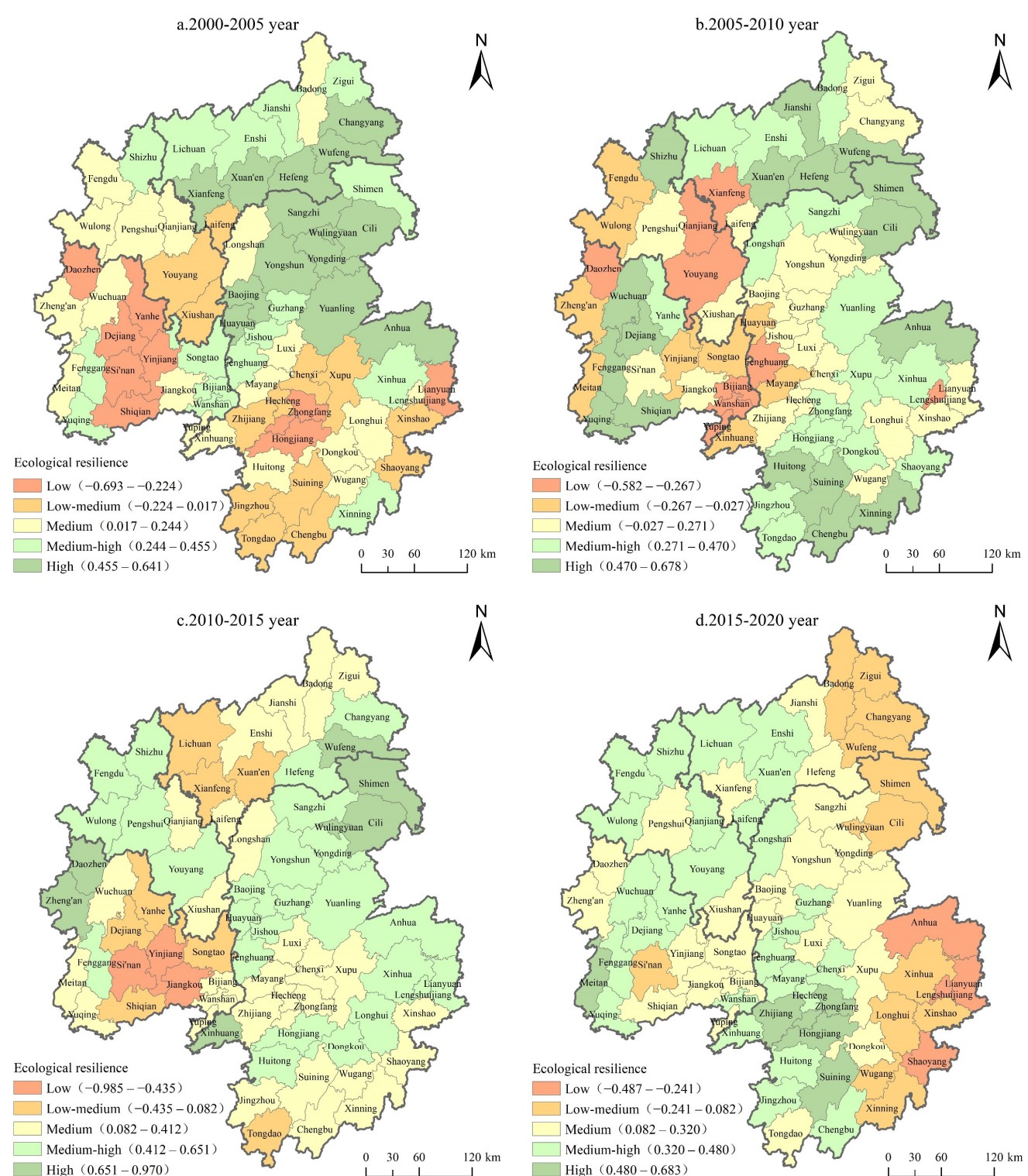

**Figure 4.** Spatial Evolution Characteristics of Ecological Resilience in the Wuling Mountains Area and its Four Sub-Regions from 2000 to 2020.

Between 2010 and 2015, areas with medium–high and high ecological resilience in the Wuling Mountains sub-region were primarily located in the northern Chongqing sub-region, western Guizhou sub-region, northeastern Hunan sub-region, and eastern Hubei sub-region. Areas with low and low–medium ecological resilience were predominantly found in the southwestern section of the Hubei sub-region and the central part of the

Guizhou sub-region. The number of counties with medium ecological resilience was the largest, the ecological resilience of these counties is mainly transformed into medium–high ecological resilience.

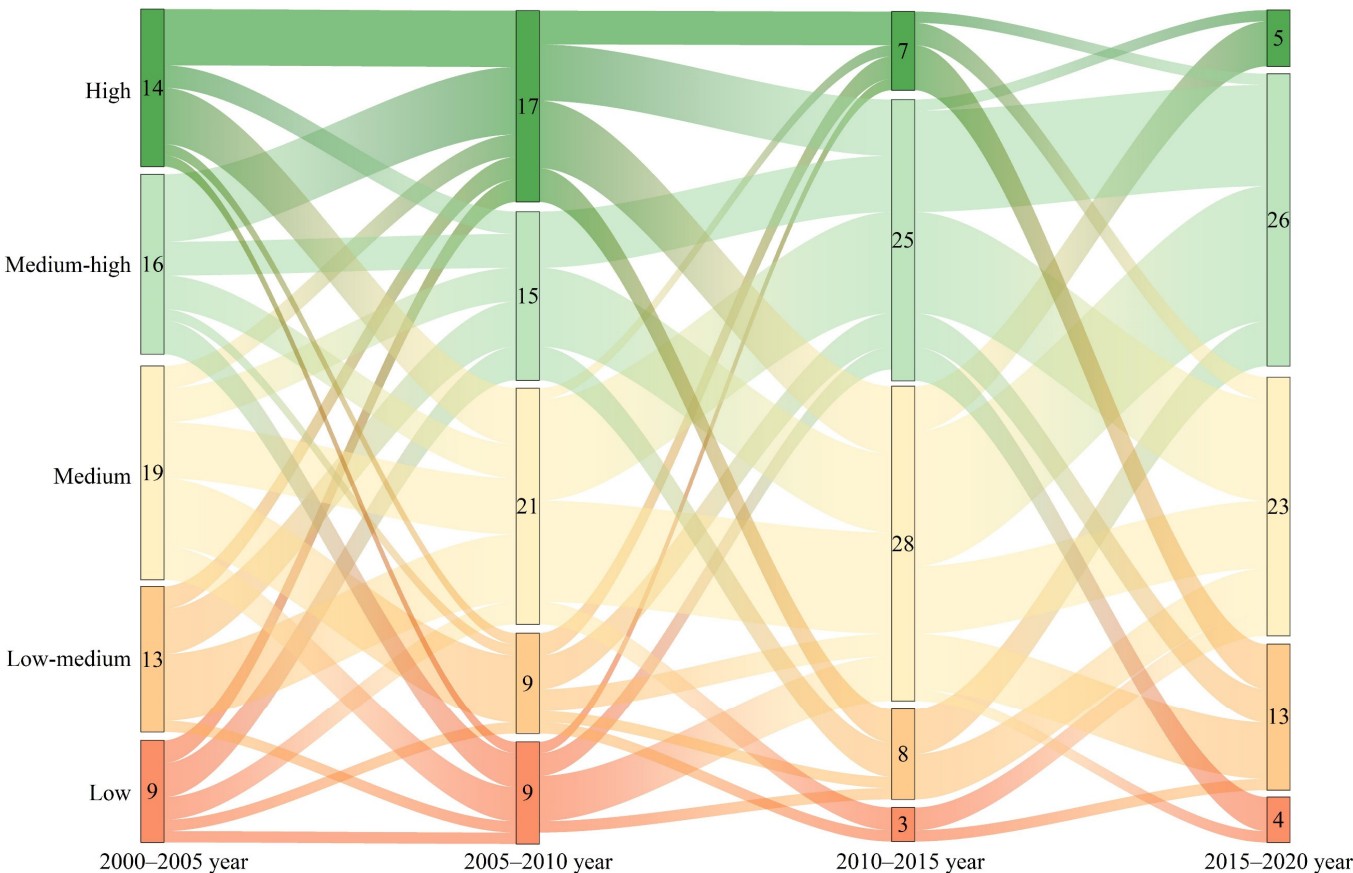

**Figure 5.** Changes in Ecological Resilience Levels in the Wuling Mountains Area from 2000 to 2020.

Between 2015 and 2020, places with medium–high and high ecological resilience in the Wuling Mountains sub-region were predominantly located in the western portion of Hunan and Hubei, the central and northern part of Chongqing, and the central part of Guizhou. Areas with low and low–medium ecological resilience were predominantly located in the eastern section of the Hunan and Hubei sub-regions. The number of counties with medium–high ecological resilience was the largest.

The ecological resilience in the Guizhou Province sub-region has shifted significantly, as shown in Figure 4. Counties with low ecological resilience shifted from the central part to the east and central parts. Those with medium–low ecological resilience shifted from the east and west to the central part. Counties with medium ecological resilience expanded from the west to the east. Those with medium–high ecological resilience were concentrated in the east and in the west. Counties with high ecological resilience decreased and shifted from the central part toward the west.

Ecological resilience in the Chongqing City sub-region showed slight variations, with low values in the southern area and high values in the northern area. Counties with low ecological resilience only occurred from 2000 to 2005 and were located in the central part. Counties with medium–low ecological resilience were observed only from 2000 to 2005 and from 2005 to 2010, and shifts in medium–low ecological resilience occurred from the south to the north. Areas with medium ecological resilience shifted from the north to the central and southern parts, and those with medium–high ecological resilience expanded from the north to the south. Counties with high ecological resilience were only found in the northern region between 2005 and 2010.

The ecological resilience in the Hubei Province sub-region shifted notably. Counties with low ecological resilience were solely found in the southwest between 2005 and 2010. Counties with medium–low ecological resilience expanded from the south to the east. Counties with medium ecological resilience expanded from the north to the central and southern parts. Counties with medium–high ecological resilience shifted from the north to the south, and counties with high ecological resilience were concentrated in the east and south.

The ecological resilience in the Hunan Province sub-region shifted significantly. Counties with low ecological resilience shifted gradually from the eastern and central-western parts to the east. Counties with medium–low ecological resilience changed from the central-southern and eastern parts to the east. Counties with medium ecological resilience shifted from the central-southern parts to the north, and counties with medium–high ecological resilience changed from a scattered distribution to the southwest. Counties with high ecological resilience shifted from the north to the south.

### 3.3. Factors Influencing Ecological Resilience in the Wuling Mountains Area

The influencing factors of ecological resilience in the Wuling Mountains area are shown in Figure 6. Between 2000 and 2020, the average values of the factors impacting the spatial and temporal distribution of ecological resilience in the Wuling Mountains area decreased in the following order: annual average temperature (0.140), annual precipitation (0.125), carbon emissions (0.114), population density (0.109), slope (0.086), elevation (0.076), GDP (0.071), and the proportion of construction land (0.063). The annual average temperature, slope, annual precipitation, proportion of construction land, elevation, and carbon emissions showed a decreasing trend in their influence on ecological resilience in the region, whereas population density and GDP exhibited an increasing trend. Minimal spatial–temporal variation occurred in the natural factors in the Wuling Mountains area during the study area. The influence of natural forces on ecological resilience was minimal. The reduced impact of building land percentage and carbon emissions on ecological resilience is due to stringent land development rules and efforts to regulate carbon emissions in the area. However, the resource problem brought by population density and population agglomeration in the Wuling Mountains area enhances the influence of population density on the ecological resilience of the area. Additionally, the Wuling Mountains area experienced a crucial phase of economic development from 2000 to 2020, resulting in increased economic growth and environmental impacts, thereby strengthening the influence of GDP on ecological resilience in the region.

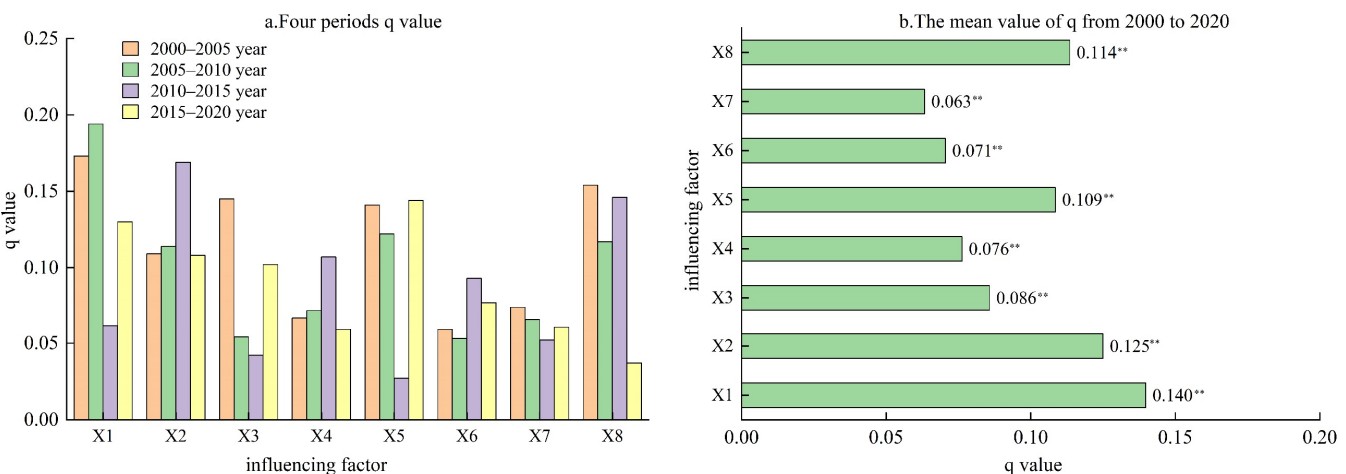

**Figure 6.** Explanatory power (q value) of influencing factors on ecological resilience in the Wuling Mountains area. X1–X8 represent the influencing factors, as described in Table 1. ** indicates a significance level of $p < 0.01$.

The spatio-temporal variation in ecological resilience in the sub-regions was mainly affected by socio-economic factors (Figures 7–10). The average value of the contributing elements determining the spatial and temporal distribution of ecological resilience in the Guizhou Province sub-region ranged from high to low as follows: the proportion of construction land (0.335), GDP (0.253), population density (0.175), carbon emissions (0.171), annual precipitation (0.113), elevation (0.035), annual temperature (0.026), and slope (0.011). Annual temperature, elevation, population density, and GDP showed an increasing trend in their influence on ecological resilience. Annual precipitation, slope, the proportion of construction land, and carbon emissions exhibit a decreasing trend. The slope factor was not statistically significant from 2010 to 2015. The trends in the influence of annual temperature and elevation on ecological resilience in the Guizhou Province sub-region were opposite to those in the Wuling Mountains area. The yearly temperature in the Guizhou Province sub-region has been increasing due to global warming, which is negatively impacting the biological ecosystem. Therefore, the influence of the annual temperature on ecological resilience in the Guizhou Province sub-region increased. Advancements in science and technology, the development, and construction can be located in areas with high elevation, but the development and construction in these areas increased the pressure of regional ecological environment. Therefore, the influence of elevation on the ecological resilience of the Guizhou Province sub-region has increased.

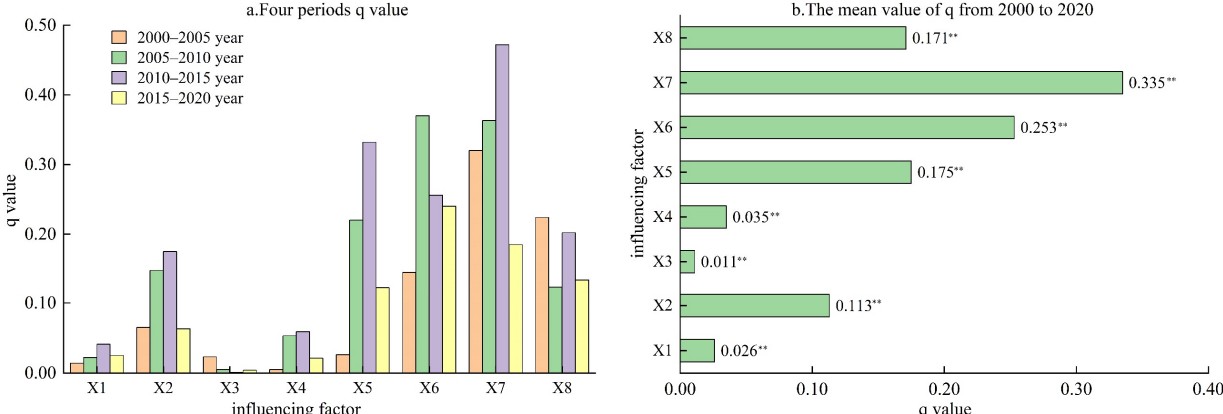

**Figure 7.** Explanatory power (q value) of influencing factors on ecological resilience in the Guizhou Province sub-region. X1–X8 represent the influencing factors, as listed in Table 1. ** indicates a significance level of $p < 0.01$.

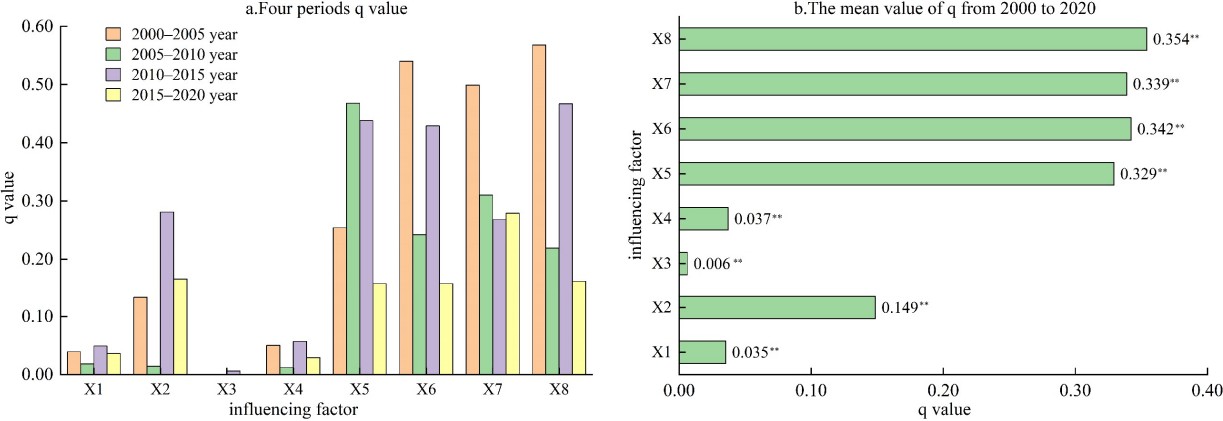

**Figure 8.** The explanatory power (q value) of influencing factors on ecological resilience in the Chongqing City sub-region. X1–X8 represent the influencing factors, as listed in Table 1. ** indicates a significance level of $p < 0.01$.

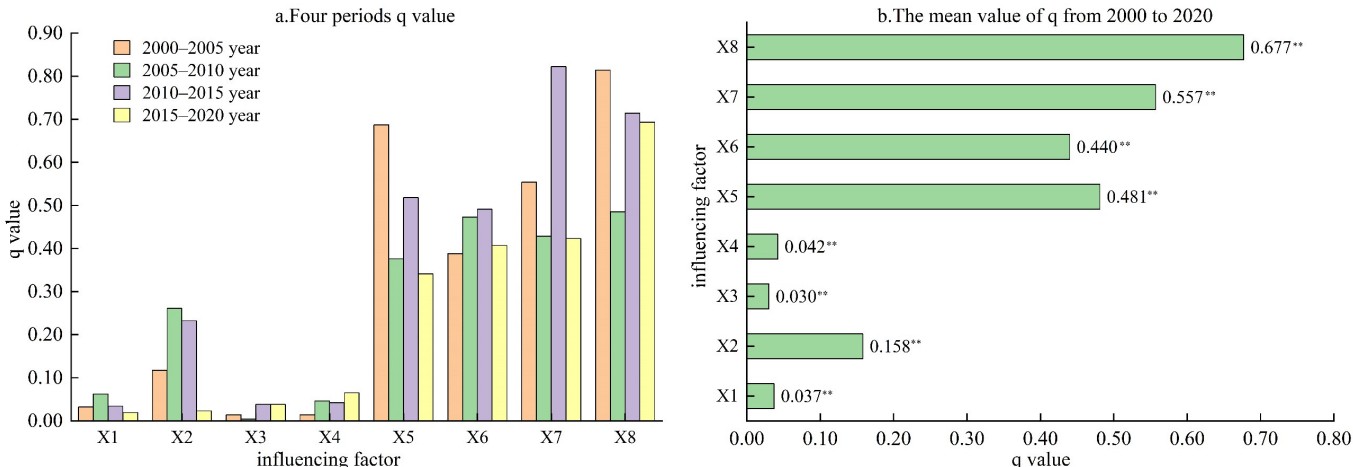

**Figure 9.** The explanatory power (q value) of the influencing factors on the ecological resilience in the Hubei Province sub-region. X1–X8 represent the influencing factors, see Table 1. ** indicates a significance level of $p < 0.01$.

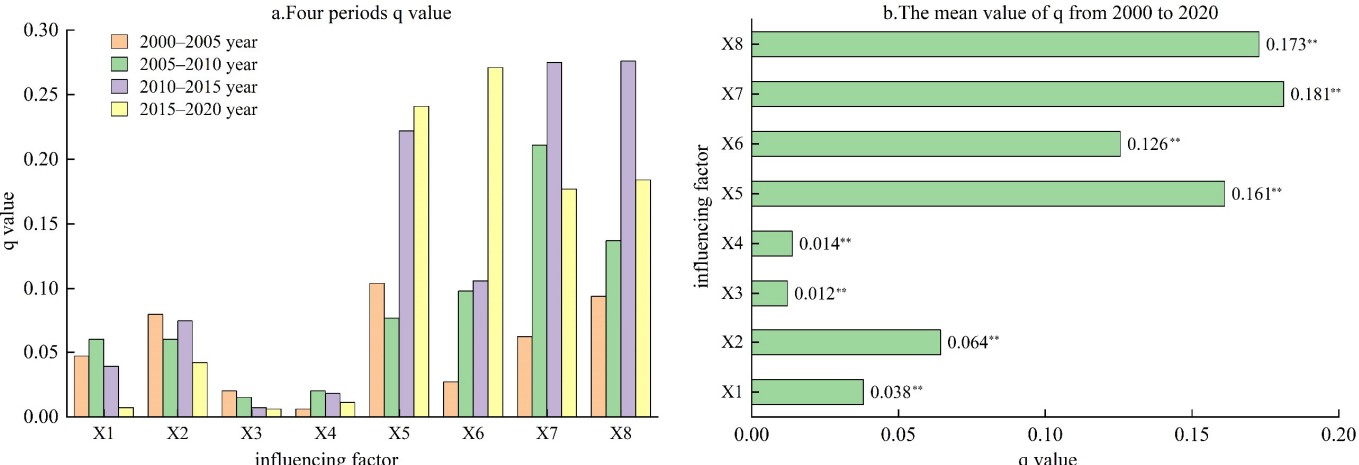

**Figure 10.** The explanatory power (q value) of the influencing factors of the Hunan Province sub-region on ecological resilience. X1–X8 represent the influencing factors, as shown in Table 1. ** indicates a significance level of $p < 0.01$.

The average values of the contributing elements of the spatial and temporal distribution of ecological resilience in the Chongqing City sub-region were ranked from large to small as follows: carbon emissions (0.354), GDP (0.342), proportion of construction land (0.339), population density (0.329), annual precipitation (0.149), elevation (0.037), annual mean temperature (0.035), and slope (0.006). The influence of annual mean temperature, population density, elevation, proportion of construction land, GDP, and carbon emissions on ecological resilience exhibited a declining pattern. Conversely, the impact of yearly rainfall on ecological resilience exhibited a rising pattern. The slope was not statistically significant in 2000–2005, 2005–2010, and 2015–2020. The trends of annual precipitation, population density, and GDP on ecological resilience in the Chongqing City sub-region were opposite to those in the Wuling Mountains area. Global warming has led to increased extreme precipitation occurrences and flood disasters in the Chongqing City sub-region, resulting in environmental damage and a greater impact of yearly precipitation on ecological resilience. Despite the high population density in the Chongqing City sub-region, efforts to reduce energy consumption and emissions mitigated the adverse effect on the environment. Hence, the influence of population density on ecological resilience in the Chongqing City sub-region decreased over time. Additionally, the promotion of sustainable economic

development in Chongqing helped mitigate the adverse effects of economic growth on the environment, leading to a decreased influence of GDP on ecological resilience.

The average value of the influencing elements affecting the spatial and temporal distribution of ecological resilience in the Hubei Province sub-region decreased from large to small in the following order: carbon emissions (0.677), proportion of construction land (0.557), population density (0.481), GDP (0.440), annual precipitation (0.158), elevation (0.042), annual average temperature (0.037), and slope (0.030). An analysis revealed a declining trend in the influence of annual average temperature, annual precipitation, population density, fraction of construction land area, and carbon emissions on ecological resilience in the region. Conversely, the impact of slope, elevation, and GDP exhibited an increasing trend. The slope factor was not statistically significant from 2005 to 2010. The changing trend of the slope's impact on ecological resilience in the Hubei Province sub-region was the opposite of that of the Wuling Mountains area. Due to scientific and technological advancements, the scope of development and construction can be expanded to areas with steeper slopes. Development and construction in these areas still exert a detrimental influence on the ecological environment. The slope's effect on ecological resilience in the Hubei Province sub-region grew stronger as time passed.

The average value of the contributing elements of the spatial and temporal distribution of ecological resilience in the Hunan Province sub-region decreased from large to small in the following order: land proportion of construction land (0.181), carbon emissions (0.173), population density (0.161), GDP (0.126), annual precipitation (0.064), average annual temperature (0.038), elevation (0.014), and slope (0.012). The influences of average annual temperature, annual precipitation, and slope on ecological resilience in the area showed a declining pattern, whereas elevation, population density, proportion of construction land, GDP, and carbon emissions demonstrated a rising pattern. The patterns of construction land proportion and carbon emissions in the Hunan Province sub-region were contrary to those in the Wuling Mountains area. The impact of the proportion of construction land and carbon emissions on the ecological resilience of the Hunan Province sub-region increased, suggesting that inadequate supervision negatively impacted environmental quality. Between 2015 and 2020, Hunan implemented stricter land -use management and carbon emission rules. This resulted in a significant decrease in the effects of construction land and carbon emissions on ecological resilience in the Hunan Province sub-region.

## 4. Discussion

The Wuling Mountains area is both ecologically vulnerable and serves as a crucial ecological function area. This paper examines the spatial and temporal evolution of ecological resilience in the Wuling Mountains area, analyzing the influencing factors at both the area and sub-area levels. The research is new in its viewpoints and focus. This research introduces novel methods for assessing ecological resilience in the Wuling Mountains and similar ecologically vulnerable regions, offering useful insights for ecological conservation and rehabilitation. Most of the existing findings measure ecological resilience by constructing a comprehensive indicator system [31,32,47,48,56,57], which requires more data and is less operational for ecological resilience studies at long time scales. In this paper, NDVI is used as a proxy for ecological resilience, and the ecological resilience index is measured using NDVI data from 2000 to 2020, which enriches the case of ecological resilience research in ecologically fragile areas and expands the scope of ecological resilience proxy selection.

The Wuling Mountains region is experiencing ecological and environmental issues like soil erosion, vegetation degradation, and frequent landslides and mudslides. The study revealed that the ecological resilience of the Wuling Mountains area between 2000 and 2020 was primarily characterized by medium and medium–high levels [15], and the growth of ecological resilience was slow and fluctuating [58], with a general upward trend [57]. The ecological resilience of the Wuling Mountains area shows obvious spatial heterogeneity and spatial agglomeration, which is consistent with the spatial distribution characteristics of natural ecological resilience in China [59]. The low and low–medium ecological resilience

areas in the Wuling Mountains area are shifted to the east, and the medium–high and high ecological resilience areas are shifted to the west and south. However, the very low ecological resilience and very high ecological resilience areas in the Dianchi Basin moved to the northeast, the light ecological resilience areas to the northwest, and the high ecological resilience areas to the southeast [31].

The impact of average yearly temperature and yearly precipitation on ecological resilience in the Wuling Mountains region was more pronounced compared to other parameters. Research has demonstrated that annual precipitation is the primary factor affecting the ecological resilience of the upper Yellow River [47]. It indicates that climatic factors exert a significant impact on ecological resilience. The effects of population density and GDP on ecological resilience in the Wuling Mountains area show an increasing trend. Population density and economic development have been found to have a negative impact on ecological resilience [57], and ecological resilience is significantly lower in areas with high population density [48]. Therefore, the Wuling Mountains region should actively address both the impact of climatic factors on ecological resilience and the impact of population and economic development on ecological resilience.

Differences exist in the elements influencing ecological resilience between the main area and its sub-areas, with socio-economic factors being the primary influencers of ecological resilience in the four sub-areas. The main influences on ecological resilience vary from region to region and county to county, depending on their development status [56]. Therefore, ecological protection and restoration programs should be tailored to local conditions. Carbon emissions were the crucial factor affecting ecological resilience in the four sub-regions, and after breakpoints, $CO_2$ was the most significant factor affecting ecosystem resilience in China [49]. Therefore, the Wuling Mountains area should strictly control carbon emissions and mitigate the negative impact on the ecological environment.

This study also has some limitations. Firstly, NDVI not only reflects ecological changes but is also sensitive to changes in land use types, and the selection of NDVI as a proxy for ecological resilience has some limitations. In future studies, we will try to use multiple nonparametric regression algorithms to measure ecological resilience and improve the scientific validity of the study. Secondly, in this paper, the study utilized LUCC data with a 250 m resolution, potentially impacting the research outcomes. In future research, we will use 30 m resolution LUCC data for the study to improve the research accuracy. Finally, the ratio of industrial output to GDP, industrial domestic solid waste, wastewater, and exhaust emissions also have an impact on ecological resilience, but these factors were not included in the indicators of ecological resilience influencing factors due to limited access to data. In our upcoming research, we will integrate the mentioned indicators into the factors influencing ecological resilience and categorize them into two dimensions: social and economic factors. This will allow for a more comprehensive analysis of the factors impacting changes in ecological resilience.

## 5. Conclusions

This paper utilized NDVI as a proxy to assess ecological resilience. The ecological resilience index was calculated based on NDVI data spanning from 2000 to 2020. The study analyzed the spatial and temporal variability of ecological resilience in the Wuling Mountains area and investigated the factors influencing ecological resilience using a geodetector. The main conclusions are as follows: (1) The growth of ecological resilience in the Wuling Mountains sub-region fluctuated slowly, with an overall upward trend. In addition to the Hubei sub-region, the ecological resilience indices of the three sub-regions of Guizhou, Chongqing, and Hunan also showed an overall upward trend. (2) The ecological resilience of the Wuling Mountains area is dominated by medium and medium–high levels. The spatial distribution of the ecological resilience index in the Wuling Mountains area and the three sub-areas of Guizhou, Hunan, and Hubei varies considerably, while changes in the Chongqing sub-area are relatively small. (3) Differences exist in the elements that affect the ecological resilience of the Wuling Mountains area and its four sub-areas. Climatic factors

like average yearly temperature and annual precipitation significantly impact the ecological resilience of the Wuling Mountains area, and population density, GDP, the proportion of built-up land area, and carbon emissions have a significant influence on the ecological resilience of the four sub-areas.

Suggestions for enhancing ecological resilience in the Wuling Mountains area are provided based on the research findings on affecting factors.

(1) The Wuling Mountains region should strengthen the monitoring and prediction of climate change and formulate climate adaptation and ecological restoration strategies.
(2) The Wuling Mountains area should formulate scientific urban planning to optimize the population layout, strengthen the public's awareness of environmental protection, and work together to maintain the ecological environment.
(3) The Wuling Mountains area should adjust its economic structure to promote industrial transformation, develop a green economy and a low-carbon economy, and promote environmental sustainability in the region.
(4) When producing construction land in the Wuling Mountains area, it is important to prioritize the protection and restoration of the ecological environment to minimize harm.
(5) The Wuling Mountains region should promote technological innovation and clean energy to reduce carbon emissions.

**Author Contributions:** Conceptualization, M.Y.; methodology, M.Y. and Y.Y.; software, J.Z. and N.Y.; formal analysis, S.L. and W.X.; writing—original draft preparation, M.Y.; writing—review and editing, M.Y., J.W., and S.L.; visualization, M.Y.; supervision, S.L., W.X. and J.W.; funding acquisition, J.W. All authors have read and agreed to the published version of the manuscript.

**Funding:** This research was funded by the National Natural Science Foundation (Grant Number: 42061036) and the Natural Science Foundation of Hunan Province (Grant Number: 2023JJ30491).

**Institutional Review Board Statement:** Not applicable.

**Informed Consent Statement:** Not applicable.

**Data Availability Statement:** Data is contained within the article.

**Acknowledgments:** We thank the anonymous reviewers for their constructive feedback.

**Conflicts of Interest:** The authors declare no conflicts of interest.

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
