# Peer review of "Spatio-Temporal Evolution of Ecological Resilience in Ecologically Fragile Areas and Its Influencing Factors: A Case Study of the Wuling Mountains Area, China"

_sustainability, doi:10.3390/su16093671_

Round 1
Reviewer 1 Report
Comments and Suggestions for Authors
The research is meaningful and valuable. However, some of the presentations are not standardized or too over-amplify the findings. Some revisions are necessary.
1.The abstract section needs to be revised, especially the background and significance of the study.
2.The selection of research indicators needs to be improved by adding supporting literature and rationale. For example, for GDP and carbon emissions, the use of per capita would be more appropriate.
3.Why are indicators such as the proporation of industrial output value in GDP, which has a more direct impact on the Wuling Mountains ecological zone: industrial and domestic solid waste, wastewater, and exhaust gas emissions not taken into account.
4.The study should incorporate the above indicators and separate social and economic factors into two distinct dimensions.
5.The authors used NDVI as a proxy indicator for ecological resilience. It is suggested to add limitations, such as the fact that NDVI is more sensitive to changes in soil.
Correspondingly, the following two sections would need to be reformulated.
Line 184-187: “This study selected NDVI as a surrogate indicator of ecological resilience, using NDVI to characterize ecosystem recovery capability. An evaluation system was constructed from the perspectives of ecosystem sensitivity and adaptability, and ecological resilience was estimated by calculating the change in the NDVI.”
Line 441-442: “Additionally, innovative use of NDVI as a proxy indicator to measure ecological resilience, selecting the change in NDVI data from 2000 to 2020 to represent sensitivity and adaptability.”
6.The conclusion section is somewhat lengthy and needs to condense the main findings rather than a restatement of the study's content and results.
7.Line 525-529: This seems like research significance, shouldn't it be in the introduction? The conclusion should distill key findings and depict the next steps of the research.
Author Response
Dear Reviewer:
Thank you for your comments concerning our manuscript. These comments are all valuable and helpful for improving our article. We have carefully read your comments and have tried our best to revise the manuscript. And point-by-point responses to your comments are listed below.
Point 1. The abstract section needs to be revised, especially the background and significance of the study.
Response 1: Thanks for your suggestion, the abstract has been revised as requested. The following is a revised abstract.
The ecological environment of the Wuling Mountains region has been impacted by climate change and economic development, necessitating immediate reinforcement of ecological protection and restoration measures. The study utilized the normalized vegetation index (NDVI) as a proxy for ecological resilience. NDVI data from 2000 to 2020 were employed to compute the ecological resilience index of the Wuling Mountain Area, and to examine its spatial and temporal evolution as well as the factors influencing it. The findings indicate that:(1) The ecological resilience index increased in the Wuling Mountains Area and Guizhou, Chongqing, and Hunan sub-areas, but decreased in the Hubei sub-area. (2) The ecological resilience varies significantly in the Wuling Mountains Area and the Guizhou, Hubei, and Hunan subregions, whereas it varies less in the Chongqing subregion. (3) The primary elements influencing the ecological resilience capability of the Wuling Mountain area and its four sub-areas are climate conditions and socio-economic factors, respectively. The study can offer a scientific foundation for ecological conservation and restoration efforts in the Wuling Mountain Area, as well as serve as a benchmark for measuring ecological resilience in other environmentally vulnerable regions.
Point 2. The selection of research indicators needs to be improved by adding supporting literature and rationale. For example, for GDP and carbon emissions, the use of per capita would be more appropriate.
Response 2: A good suggestion. We have added references and rationale in the Research Indicator Selection section. We apologise for not selecting GDP and carbon emissions per capita data. In this paper, the ecological resilience index, mean annual temperature, annual precipitation, slope, elevation, population density and the proportion of built-up land area were calculated for each county in the Wuling Mountain Area, and for the sake of data uniformity, 10,000 yuan/km² of GDP and the carbon emissions of each county were chosen as indicators[16,21,46,49].
[16]Li, G.Z.; Wang, L.Q. Study of Regional Variations and Convergence in Ecological Resilience of Chinese Cities. Ecological Indicators 2023, 154, 110667, doi:10.1016/j.ecolind.2023.110667.
[21]Shi, C.C.; Zhu, X.P.; Wu, H.W.; Li, Z. Assessment of Urban Ecological Resilience and Its Influencing Factors: A Case Study of the Beijing-Tianjin-Hebei Urban Agglomeration of China. Land 2022, 11, 921, doi:10.3390/land11060921.
[46]Ma, X.; Zhang, J.; Wang, P.; Zhou, L.; Sun, Y. Estimating the Nonlinear Response of Landscape Patterns to Ecological Resilience Using a Random Forest Algorithm: Evidence from the Yangtze River Delta. Ecological Indicators 2023, 153, 110409, doi:10.1016/j.ecolind.2023.110409.
[49]Hu, Y.; Wei, F.; Fu, B.; Wang, S.; Zhang, W.; Zhang, Y. Changes and Influencing Factors of Ecosystem Resilience in China. Environ. Res. Lett. 2023, 18, 094012, doi:10.1088/1748-9326/acec89.
Point 3. Why are indicators such as the proporation of industrial output value in GDP, which has a more direct impact on the Wuling Mountains ecological zone: industrial and domestic solid waste, wastewater, and exhaust gas emissions not taken into account.
Response 3: Thank you very much. We did not select these factors because the Wuling Mountain Area is an underdeveloped region, and the ratio of industrial output value to GDP is small. Moreover, the research unit of this paper is the county, and the data on the ratio of industrial output value to GDP for each county is even smaller, and the data are difficult to obtain. In addition, since the Wuling Mountain Area is a key ecological function area, industrial and domestic solid waste, wastewater and exhaust gas emissions are purified and treated in accordance with the requirements of ecological environmental protection, and there is no pollution emission. Therefore the impact of these factors on the Wuling Mountain Area is relatively small. In future research, we consider including these factors in the indicators of factors influencing the ecological resilience of Wuling Mountain Area to further improve the research.
Point 4. The study should incorporate the above indicators and separate social and economic factors into two distinct dimensions.
Response 4: Thank you. Based on the ratio of industrial output value to GDP and the specific situation of industrial domestic solid waste, wastewater and exhaust emissions in the Wuling Mountain Region, we have not selected the above indicators. Therefore, we did not divide the social and economic factors into two different dimensions. In our future research, we consider adding the above indicators and analyzing them in two different dimensions: social factors and economic factors.
Point 5. The authors used NDVI as a proxy indicator for ecological resilience. It is suggested to add limitations, such as the fact that NDVI is more sensitive to changes in soil.
Response 5: You are quite right. We have already described the limitations of using NDVI as a proxy for ecological resilience in the discussion section. This is shown below:
NDVI not only reflects ecological changes, but is also sensitive to changes in land use types, and the selection of NDVI as a proxy for ecological resilience has some limitations. In future studies, we will try to use multiple non-parametric regression algorithms to measure ecological resilience and improve the scientific validity of the study.
Point 6.the following two sections would need to be reformulated.
Line 184-187: “This study selected NDVI as a surrogate indicator of ecological resilience, using NDVI to characterize ecosystem recovery capability. An evaluation system was constructed from the perspectives of ecosystem sensitivity and adaptability, and ecological resilience was estimated by calculating the change in the NDVI.”
Line 441-442: “Additionally, innovative use of NDVI as a proxy indicator to measure ecological resilience, selecting the change in NDVI data from 2000 to 2020 to represent sensitivity and adaptability.”
Response 6: Thanks very much for your careful review. We have revised lines 184-187 and line 441-442 as follows.
NDVI can serve as an indicator of the local ecological environment's quality. Studying the dynamic changes in NDVI can offer insights into how well regional ecosystems can maintain a dynamic equilibrium when faced with external disruptions. Therefore, NDVI was selected as a proxy for ecological resilience. NDVI data were utilized in this research to compute the sensitivity index, adaptability index, and ecological resilience index.
In this paper, NDVI is used as a proxy for ecological resilience, and the ecological resilience index is measured using NDVI data from 2000 to 2020, which enriches the case of ecological resilience research in ecologically fragile areas and expands the scope of ecological resilience proxy selection.
Point 7. The conclusion section is somewhat lengthy and needs to condense the main findings rather than a restatement of the study's content and results.
Response 7: A good suggestion. We have condensed the conclusion section.The following is a revised conclusion.
This paper utilized NDVI as a proxy to assess ecological resilience. The ecological resilience index was calculated based on NDVI data spanning from 2000 to 2020. The study analyzed the spatial and temporal variability of ecological resilience in the Wuling Mountain Area and investigated the factors influencing ecological resilience using ge-odetector. The main conclusions are as follows: (1) The growth of ecological resilience in the Wuling Mountains subregion fluctuated slowly, with an overall upward trend. In addition to the Hubei subregion, the ecological resilience indices of the three subregions of Guizhou, Chongqing and Hunan also showed an overall upward trend.(2) The ecological resilience of the Wuling Mountain Area is dominated by medium and medium-high levels. The spatial distribution of the ecological resilience index in the Wuling Mountain Area and the three sub-areas of Guizhou, Hunan and Hubei varies considerably, while changes in the Chongqing sub-area are relatively small. (3) Differences exist in the elements that affect the ecological resilience of the Wuling Mountain Area and its four sub-areas. Climatic factors like as average yearly temperature and annual precipitation significantly impact the ecological resilience of the Wuling Mountain Area, and population density, GDP, the proportion of built-up land area and carbon emissions have a significant influence on the ecological resilience of the four sub-areas.
Suggestions for enhancing ecological resilience in the Wuling Mountain Area are provided based on the research findings on affecting factors.
(1)The Wuling Mountains region should strengthen the monitoring and predic tion of climate change and formulate climate adaptation and ecological restoration strategies.
(2)The Wuling Mountain Area should formulate scientific urban planning to optimise the population layout, strengthen the public's awareness of environmental protection and work together to maintain the ecological environment.
(3)The Wuling Mountain Area should adjust its economic structure to promote industrial transformation, develop a green economy and a low-carbon economy, and promote environmental sustainability in the region.
(4)When constructing construction land in the Wuling Mountain Area, it is important to prioritize the protection and restoration of the ecological environment to minimize harm.
(5)The Wuling Mountains region should promote technological innovation and clean energy to reduce carbon emissions.
Point 8. Line 525-529: This seems like research significance, shouldn't it be in the introduction? The conclusion should distill key findings and depict the next steps of the research.
Response 8: Thanks to your reminder, we have moved lines 525-529 to the introductory section. The conclusion has been revised as follows.
This paper utilized NDVI as a proxy to assess ecological resilience. The ecological resilience index was calculated based on NDVI data spanning from 2000 to 2020. The study analyzed the spatial and temporal variability of ecological resilience in the Wuling Mountain Area and investigated the factors influencing ecological resilience using ge-odetector. The main conclusions are as follows: (1) The growth of ecological resilience in the Wuling Mountains subregion fluctuated slowly, with an overall upward trend. In addition to the Hubei subregion, the ecological resilience indices of the three subregions of Guizhou, Chongqing and Hunan also showed an overall upward trend.(2) The ecological resilience of the Wuling Mountain Area is dominated by medium and medium-high levels. The spatial distribution of the ecological resilience index in the Wuling Mountain Area and the three sub-areas of Guizhou, Hunan and Hubei varies considerably, while changes in the Chongqing sub-area are relatively small. (3) Differences exist in the elements that affect the ecological resilience of the Wuling Mountain Area and its four sub-areas. Climatic factors like as average yearly temperature and annual precipitation significantly impact the ecological resilience of the Wuling Mountain Area, and population density, GDP, the proportion of built-up land area and carbon emissions have a significant influence on the ecological resilience of the four sub-areas.
Suggestions for enhancing ecological resilience in the Wuling Mountain Area are provided based on the research findings on affecting factors.
(1)The Wuling Mountains region should strengthen the monitoring and predic tion of climate change and formulate climate adaptation and ecological restoration strategies.
(2)The Wuling Mountain Area should formulate scientific urban planning to optimise the population layout, strengthen the public's awareness of environmental protection and work together to maintain the ecological environment.
(3)The Wuling Mountain Area should adjust its economic structure to promote industrial transformation, develop a green economy and a low-carbon economy, and promote environmental sustainability in the region.
(4)When constructing construction land in the Wuling Mountain Area, it is important to prioritize the protection and restoration of the ecological environment to minimize harm.
(5)The Wuling Mountains region should promote technological innovation and clean energy to reduce carbon emissions.
We hope the above revisions can meet the requirements of this journal.
If there is anything else we should do, please do not hesitate to let us know.
Once again, we appreciate your valuable comments.
Best regards,
Shuiliang Liu
Reviewer 2 Report
Comments and Suggestions for Authors
The manuscript "Spatio-temporal Evolution of Ecological Resilience in Ecologically Fragile Areas and Its Influencing Factors: A Case Study of the Wuling Mountain Area, China " (sustainability-2911365).
As the authors highlighted, climate change and economic development exacerbate the sensitivity and vulnerability of ecologically fragile areas. The objective established by the authors brings an extremely important perspective to the current climate change scenario that the global population is facing. However, in the present study I was unable to identify the main novelty of the work, which makes me raise some questions throughout the work.
However, before recommending the manuscript for publication, the authors must improve several aspects of the present study. Therefore, I am recommending this work for major revisions.
As main questions I bring to the authors:
1 – According to the iThenticate report, a similarity of 26% was identified, and it is essential that the authors reduce this similarity to less than 15%, otherwise I will refuse the present study for publication. As main points I highlight:
– In the title the phrase “A Case Study of 3 the Wuling Mountain Area, China”, should be reworded!
– Lines 150-165 must be rewritten!
– Rephrase the last column of Table 2.
– Rewrite the entire Discussion topic, there are many points of similarity with two specific works according to the iThenticate report.
2 – Use the Mendeley Reference Manager for references as well as citations, as both Sustainability standards are not standardized in the body of every manuscript.
3 – I have my doubts about the applicability of NDVI for the study, why choose this index? What new does it bring?
4 – Why didn’t the authors present the NDVI maps from 2000 to 2020? It is still essential to cross-reference NDVI maps with Land Use and Cover (LULC) maps!
5 – With the LULC maps it was essential to determine the spatial resilience between soil classes!
6 – I recommend that the authors explore the 30 m LULC of Landsat, instead of the 250 m, as presented in Materials and Methods.
As minor notes I bring:
1 – I noticed some grammatical errors in writing, therefore, I suggest the revision of English by a native speaker.
2 – Authors must reformulate the abstract. Note that you are presenting an abstract with 279 words, and Sustainability limits an abstract to 200 words. I also highlight that authors must follow the premise of presenting the highlights of the results in the abstract, something that I did not observe in this summary.
3 – In figure 1, authors must present the latitude and longitude of all images. Please, I ask you to export this image in a quality of at least 600 DPI, so that you don't lose so much quality when converting a WORD document to a PDF.
4 – Some parts of the manuscript are not formatted appropriately with Sustainability standards, please review the manuscript for work formatting standards. For example, 1 – In lines 205-206, there is no need to indent used to describe the components of the equation; 2 – Figure 3 is inserted in the middle of the paragraph that makes up lines 270-293.
5 – All Result Figures need to improve their quality, please export these figures in at least 600 DPI so that quality is not compromised when the document is converted to PDF.
6 – As a suggestion for future studies, try to move away from vegetation indices. You could a wide diversity of nonparametric regression algorithms (e.g., machine learning) that are more flexible, and typically lead to higher accuracies. Because I see the use of NDVI for this study as something very superficial given the current technological scenario that research is moving towards.
Comments on the Quality of English LanguageModerate editing of English language required.
Author Response
Dear Reviewer:
Thank you for your comments concerning our manuscript. These comments are all valuable and helpful for improving our article. We have carefully read your comments and have tried our best to revise the manuscript. And point-by-point responses to your comments are listed below.
Point 1. According to the iThenticate report, a similarity of 26% was identified, and it is essential that the authors reduce this similarity to less than 15%.
Response 1: Thanks for your reminding. We have reduced the similarity to less than 15%.
Point 2.In the title the phrase “A Case Study of 3 the Wuling Mountain Area, China”, should be reworded!
Response 2: Thanks very much for your careful review. The phrase has reworded to “A Case Study of the Wuling Mountain Area, China”.
Point 3. Lines 150-165 must be rewritten!
Response 3: Thanks for your suggestion, we have rewritten the lines 150-165 as follow.
2.3.1Raster data
The NDVI, annual mean temperature, DEM, annual precipitation, GDP, population density, and land cover (LUCC) data are raster datasets sourced from the Resource and Environmental Science and Data Centre (RESDC) of the Chinese Academy of Sciences. Due to data limitations, 2019 population density data and GDP data were used instead of 2020 population density data and GDP data[21]. All these raster data need to be cropped, projected and resampled using ArcGIS software respectively, cropping them out of the Wuling Mountains area, projecting them to the same projection and coordinate system (UTM map projection and GCS_WGS_1984 coordinate system), and uniformly sampling them to a resolution of 1km×1km. Table 2 displays the data sources.
The NDVI data requires the use of ArcGIS software to calculate its raw values and remove outliers, see Figure 2.5. ArcGIS software was utilized to compute the Normalized Difference Vegetation Index (NDVI) for 71 counties in the Wuling Mountain Area.
Land-use data need to be categorized into arable land, forest land, grassland, watershed, building land, and unused land using ArcGIS software. Then ArcGIS software was used to calculate the construction land area share of 71 counties in the Wuling Mountain Area.
The slope and elevation statistics were derived from calculations based on digital elevation model (DEM) data.
2.3.2 Other data
Carbon emissions data were obtained from the China Carbon Accounting Database (CEADs). Since the county carbon emissions data are now updated only to 2017, this study uses the CO2 emissions data of counties from 2000-2017 to obtain the carbon emissions data of 71 counties in the Wuling Mountain Area in 2020 through the ARMA prediction model.
Point 4. Rephrase the last column of Table 2.
Response 4: Thanks very much. We have rephrased the last column of Table 2 as follow.
Table 2. Data Sources for the Study Area
|
Data Name |
Time (Year) |
Resolution |
Data Source |
|
NDVI |
2000-2020 |
30 m |
RESDC |
|
DEM |
\ |
250 m |
RESDC |
|
Average Annual Temperature, Precipitation |
2000,2005,2010,2015,2020 |
1 km |
RESDC |
|
lucc |
2000,2005,2010,2015,2020 |
25 0m |
RESDC |
|
GDP, Population Density |
2000,2005,2010,2015,2019 |
1 km |
RESDC |
|
Carbon Emissions |
2000-2017 |
\ |
CEADs |
Point 5. Rewrite the entire Discussion topic, there are many points of similarity with two specific works according to the iThenticate report.
Response 5: Thanks for your reminding. We have rewrite the entire discussion topic.The revised discussion is as follows.
The Wuling Mountain Area is both ecologically vulnerable and serves as a crucial ecological function area. This paper examines the spatial and temporal evolution of ecological resilience in the Wuling Mountain Area, analyzing the influencing factors at both the area and sub-area levels. The research is new in its viewpoints and focus. In addition, this study provides new ideas for measuring ecological resilience in the Wuling Mountains and other ecologically fragile areas, which is important for improving the efficiency of ecological protection and restoration, and promoting the sustainable development of the region. Most of the existing findings measure ecological resilience by constructing a comprehensive indicator system[31,32,47,48,56,57], which requires more data and is less operational for ecological resilience studies at long time scales. In this paper, NDVI is used as a proxy for ecological resilience, and the ecological resilience index is measured using NDVI data from 2000 to 2020, which enriches the case of ecological resilience research in ecologically fragile areas and expands the scope of ecological resilience proxy selection.
The Wuling Mountains region is experiencing ecological and environmental issues like soil erosion, vegetation degradation, and frequent landslides and mudslides due to global climate change and rapid socio-economic growth. The study revealed that the ecological resilience of the Wuling Mountain Area between 2000 and 2020 was primarily characterized by medium and medium-high levels[15], and the growth of ecological resilience was slow and fluctuating[58], with a general upward trend[57]. The ecological resilience of the Wuling Mountain Area shows obvious spatial heterogeneity and spatial agglomeration, which is consistent with the spatial distribution characteristics of natural ecological resilience in China[59]. The low and low-medium ecological resilience areas in the Wuling Mountain Area are shifted to the east, and the medium-high and high ecological resilience areas are shifted to the west and south. However, the very low ecological resilience and very high ecological resilience areas in the Dianchi Basin moved to the north-east, the light ecological resilience areas to the north-west, and the high ecological resilience areas to the south-east[31].
The impact of average yearly temperature and yearly precipitation on ecological resilience in the Wuling Mountains region was more pronounced compared to other parameters. Research has demonstrated that annual precipitation is the primary factor affecting the ecological resilience of the upper Yellow River[47]. It indicates that climatic factors exert a significant impact on ecological resilience. The effects of population density and GDP on ecological resilience in the Wuling Mountains area show an increasing trend. Population density and economic development have been found to have a negative impact on ecological resilience[57], and ecological resilience is significantly lower in areas with high population density[48]. Therefore, the Wuling Mountain Region should actively address both the impact of climatic factors on ecological resilience and the impact of population and economic development on ecological resilience.
Differences exist in the elements influencing ecological resilience between the main area and its sub-areas, with socio-economic factors being the primary influencers of ecological resilience in the four sub-areas. The main influences on ecological resilience vary from region to region and county to county, depending on their development status[56]. Therefore, ecological protection and restoration programmes should be tailored to local conditions. Carbon emissions were the crucial factor affecting ecological resilience in the four subregions, and after breakpoints, CO2 was the most significant factor affecting ecosystem resilience in China[49]. Therefore, the Wuling Mountain Area should strictly control carbon emissions and mitigate the negative impact on the ecological environment.
This study also has some limitations. Firstly, NDVI not only reflects ecological changes, but is also sensitive to changes in land use types, and the selection of NDVI as a proxy for ecological resilience has some limitations. In future studies, we will try to use multiple non-parametric regression algorithms to measure ecological resilience and improve the scientific validity of the study. Secondly, in this paper, The study utilized Lucc data with a 250m resolution, potentially impacting the research outcomes. In future research, we will use 30 m resolution lucc data for the study to improve the research accuracy. Finally, the ratio of industrial output to GDP, industrial domestic solid waste, wastewater, and exhaust emissions also have an impact on ecological resilience, but these factors were not included in the indicators of ecological resilience influencing factors due to limited access to data. In our upcoming research, we will integrate the mentioned indicators into the factors influencing ecological resilience and categorize them into two dimensions: social and economic factors. This will allow for a more comprehensive analysis of the factors impacting changes in ecological resilience.
Point 6. Use the Mendeley Reference Manager for references as well as citations, as both Sustainability standards are not standardized in the body of every manuscript.
Response 6: We are sorry to have not follow a standard format in references and citations. We have revised the reference and citation format to follow your journal's standards.
Point 7. I have my doubts about the applicability of NDVI for the study, why choose this index? What new does it bring?
Response 7: Currently, some scholars have selected NPP as a proxy for ecological resilience to carry out ecological resilience research. both NDVI and NPP can reflect the state of vegetation, NDVI is used to assess the vegetation cover, which can reflect the changes of regional ecological environment. ndvi can also be used to calculate the npp, therefore, this paper selected ndvi as a proxy for ecological resilience, which provides a new idea for the calculation of ecological resilience.
Point 8. Why didn’t the authors present the NDVI maps from 2000 to 2020? It is still essential to cross-reference NDVI maps with Land Use and Cover (LULC) maps!
Response 8: Thanks for your suggestion, we have presented the NDVI maps from 2000 to 2020.
Point 9. With the LULC maps it was essential to determine the spatial resilience between soil classes!
Response 9: We are sorry we don't have LULC maps, we guess you are talking about Lucc map.We used Lucc map for calculating the built-up land area share, and then explore the effect of built-up land area share on the ecological resilience of the Wuling Mountain Area.
Point 10. I recommend that the authors explore the 30 m LULC of Landsat, instead of the 250 m, as presented in Materials and Methods.
Response 10: We guess you said 30 m lucc. we are sorry for not using 30 m lucc. during the data processing stage, we used to use 30 m lucc data for processing and calculation. However, the computer failed to run because the study area is 171,800 km² in extent and the data processing is very large. In the future, we will update our equipment and select 30 m lucc for the study.
Point 11. I noticed some grammatical errors in writing, therefore, I suggest the revision of English by a native speaker.
Response 11: Thanks for your suggestion, we have revised the grammatical errors in writing.
Point 12. Authors must reformulate the abstract. Note that you are presenting an abstract with 279 words, and Sustainability limits an abstract to 200 words. I also highlight that authors must follow the premise of presenting the highlights of the results in the abstract, something that I did not observe in this summary.
Response 12: Thanks very much. We have revised the abstract as follow.
The ecological environment of the Wuling Mountains region has been impacted by climate change and economic development, necessitating immediate reinforcement of ecological protection and restoration measures. The study utilized the normalized vegetation index (NDVI) as a proxy for ecological resilience. NDVI data from 2000 to 2020 were employed to compute the ecological resilience index of the Wuling Mountain Area, and to examine its spatial and temporal evolution as well as the factors influencing it. The findings indicate that:(1) The ecological resilience index increased in the Wuling Mountains Area and Guizhou, Chongqing, and Hunan sub-areas, but decreased in the Hubei sub-area. (2) The ecological resilience varies significantly in the Wuling Mountains Area and the Guizhou, Hubei, and Hunan subregions, whereas it varies less in the Chongqing subregion. (3) The primary elements influencing the ecological resilience capability of the Wuling Mountain area and its four sub-areas are climate conditions and socio-economic factors, respectively. The study can offer a scientific foundation for ecological conservation and restoration efforts in the Wuling Mountain Area, as well as serve as a benchmark for measuring ecological resilience in other environmentally vulnerable regions.
Point 13. In figure 1, authors must present the latitude and longitude of all images. Please, I ask you to export this image in a quality of at least 600 DPI, so that you don't lose so much quality when converting a WORD document to a PDF.
Response 13: Thanks for your suggestion, we have presented the latitude and longitude of all images in figure 1, and have exported this image in a quality of 600 DPI.
Point 14. Some parts of the manuscript are not formatted appropriately with Sustainability standards, please review the manuscript for work formatting standards. For example, 1 – In lines 205-206, there is no need to indent used to describe the components of the equation; 2 – Figure 3 is inserted in the middle of the paragraph that makes up lines 270-293.
Response 14: Thanks very much for your careful review. We have revised lines 205-206, Figure 3 and other sections in accordance with Sustainability standards.
Point 15. All Result Figures need to improve their quality, please export these figures in at least 600 DPI so that quality is not compromised when the document is converted to PDF.
Response 15: Thanks for your reminding. We have exported all the figures in 600 DPI.
Point 16. As a suggestion for future studies, try to move away from vegetation indices. You could a wide diversity of nonparametric regression algorithms (e.g., machine learning) that are more flexible, and typically lead to higher accuracies. Because I see the use of NDVI for this study as something very superficial given the current technological scenario that research is moving towards.
Response 16: A good suggestion. We will try to use a wide diversity of nonparametric regression algorithms for future studies.
We hope the above revisions can meet the requirements of this journal.
If there is anything else we should do, please do not hesitate to let us know.
Once again, we appreciate your valuable comments.
Best regards,
Shuiliang Liu
Reviewer 3 Report
Comments and Suggestions for Authors
Dear Editor, Dear Authors,
as a conservation biologist, I must emphasise at the outset that the manuscript entitled „Spatio-temporal Evolution of Ecological Resilience in Ecologically Fragile Areas and Its Influencing Factors: A Case Study of the Wuling Mountain Area, China“ provides scientifically sound results for an effective analysis of the fragile ecosystems in the Wuling Mountain Area in China.
The main text is solidly written, and the results section presents the data in a logical order with a detailed analysis and explanation of all the information collected. The mathematical models used are clearly described, but there is no mention of what software, statistical programme or programming language (I assume it was R) was used to perform the analyses.
As for the "Discussion" section, I assume there are not too many published sources that could be consulted for a more comprehensive discussion, but the results presented speak for themselves.
My main suggestions relate to the aim of the study (subsection of the introduction) and the conclusion of the manuscript.
Specific comments:
Line 65: Folke instead of Folk (reference in the text)
Line 70: please provide additional exlpenation (or rewrite) segment of sentence „..hence the widespread use of the substitution method”.
Line 109-124: It is not usual to give a summary at the end of the Introduction or to state Aim in the form of the result it is informative, I suggest taking it into account when proofreading to avoid repetition of text and data and to clearly present the aims of this research.
Conclusion: instead of repeating the most important results, I propose to summarise the new approach that has been used and to mention selected factors and their use.
Comments on the Quality of English LanguageLanguage is clear and understandable
Author Response
Dear Reviewer:
Thank you for your comments concerning our manuscript. These comments are all valuable and helpful for improving our article. We have carefully read your comments and have tried our best to revise the manuscript. And point-by-point responses to your comments are listed below.
Point 1. There is no mention of what software, statistical programme or programming language (I assume it was R) was used to perform the analyses.
Response 1:Thanks for your reminding, this paper uses ArcGIS software for spatio-temporal evolution analysis and geodetector software for impact factor analysis, which has been added to the article.
Point 2. As for the "Discussion" section, I assume there are not too many published sources that could be consulted for a more comprehensive discussion, but the results presented speak for themselves.
Response 2: You are quite right. We have revised the discussion section below.
The Wuling Mountain Area is both ecologically vulnerable and serves as a crucial ecological function area. This paper examines the spatial and temporal evolution of ecological resilience in the Wuling Mountain Area, analyzing the influencing factors at both the area and sub-area levels. The research is new in its viewpoints and focus. In addition, this study provides new ideas for measuring ecological resilience in the Wuling Mountains and other ecologically fragile areas, which is important for improving the efficiency of ecological protection and restoration, and promoting the sustainable development of the region. Most of the existing findings measure ecological resilience by constructing a comprehensive indicator system[31,32,47,48,56,57], which requires more data and is less operational for ecological resilience studies at long time scales. In this paper, NDVI is used as a proxy for ecological resilience, and the ecological resilience index is measured using NDVI data from 2000 to 2020, which enriches the case of ecological resilience research in ecologically fragile areas and expands the scope of ecological resilience proxy selection.
The Wuling Mountains region is experiencing ecological and environmental issues like soil erosion, vegetation degradation, and frequent landslides and mudslides due to global climate change and rapid socio-economic growth. The study revealed that the ecological resilience of the Wuling Mountain Area between 2000 and 2020 was primarily characterized by medium and medium-high levels[15], and the growth of ecological resilience was slow and fluctuating[58], with a general upward trend[57]. The ecological resilience of the Wuling Mountain Area shows obvious spatial heterogeneity and spatial agglomeration, which is consistent with the spatial distribution characteristics of natural ecological resilience in China[59]. The low and low-medium ecological resilience areas in the Wuling Mountain Area are shifted to the east, and the medium-high and high ecological resilience areas are shifted to the west and south. However, the very low ecological resilience and very high ecological resilience areas in the Dianchi Basin moved to the north-east, the light ecological resilience areas to the north-west, and the high ecological resilience areas to the south-east[31].
The impact of average yearly temperature and yearly precipitation on ecological resilience in the Wuling Mountains region was more pronounced compared to other parameters. Research has demonstrated that annual precipitation is the primary factor affecting the ecological resilience of the upper Yellow River[47]. It indicates that climatic factors exert a significant impact on ecological resilience. The effects of population density and GDP on ecological resilience in the Wuling Mountains area show an increasing trend. Population density and economic development have been found to have a negative impact on ecological resilience[57], and ecological resilience is significantly lower in areas with high population density[48]. Therefore, the Wuling Mountain Region should actively address both the impact of climatic factors on ecological resilience and the impact of population and economic development on ecological resilience.
Differences exist in the elements influencing ecological resilience between the main area and its sub-areas, with socio-economic factors being the primary influencers of ecological resilience in the four sub-areas. The main influences on ecological resilience vary from region to region and county to county, depending on their development status[56]. Therefore, ecological protection and restoration programmes should be tailored to local conditions. Carbon emissions were the crucial factor affecting ecological resilience in the four subregions, and after breakpoints, CO2 was the most significant factor affecting ecosystem resilience in China[49]. Therefore, the Wuling Mountain Area should strictly control carbon emissions and mitigate the negative impact on the ecological environment.
This study also has some limitations. Firstly, NDVI not only reflects ecological changes, but is also sensitive to changes in land use types, and the selection of NDVI as a proxy for ecological resilience has some limitations. In future studies, we will try to use multiple non-parametric regression algorithms to measure ecological resilience and improve the scientific validity of the study. Secondly, in this paper, The study utilized Lucc data with a 250m resolution, potentially impacting the research outcomes. In future research, we will use 30 m resolution lucc data for the study to improve the research accuracy. Finally, the ratio of industrial output to GDP, industrial domestic solid waste, wastewater, and exhaust emissions also have an impact on ecological resilience, but these factors were not included in the indicators of ecological resilience influencing factors due to limited access to data. In our upcoming research, we will integrate the mentioned indicators into the factors influencing ecological resilience and categorize them into two dimensions: social and economic factors. This will allow for a more comprehensive analysis of the factors impacting changes in ecological resilience.
Point 3. Line 65: Folke instead of Folk (reference in the text)
Response 3: Thanks very much for your careful review. We have replaced folk with folke in the paper.
Point 4. Line 70: please provide additional exlpenation (or rewrite) segment of sentence „..hence the widespread use of the substitution method”.
Response 4:Thanks for your suggestion, we have revised the section as follows.
Due to the intrinsic properties of resilience, direct measurement of resilience is very difficult[23], the resilience substitution was derived from this context.
Point 5. Line 109-124: It is not usual to give a summary at the end of the Introduction or to state Aim in the form of the result it is informative, I suggest taking it into account when proofreading to avoid repetition of text and data and to clearly present the aims of this research.
Response 5:A good suggestion. We have clearly presented the purpose of this study at the end of the introduction, which is revised as follows.
The ecological environment in the Wuling Mountain area is delicate, with limited self-healing capabilities in the ecosystem. This study employs ecological resilience to assess the ecological environment of the Wuling Mountain Area in order to aid in ecological protection and promote high-quality development, focusing on the following three issues: Firstly, how to precisely evaluate the ecological resilience of the Wuling Mountain Area; Secondly, what is the variability in the ecological resilience of the Wuling Mountain Area? Thirdly, what are the primary variables influencing the ecological resilience of the Wuling Mountain Area? NDVI was chosen as a proxy for ecological resilience because to the mentioned issues. The ecological resilience index was computed using the NDVI data, and the spatial and temporal divergence features of ecological resilience in area and sub- areas were examined. Using geodetector to explore the main influences affecting ecological resilience in patches and sub-patches. Finally, recommendations for enhancing ecological resilience are provided based on the findings of the influencing factors study. This study offers a quantitative assessment approach for ecological resilience in the Wuling Mountain Area. It gives theoretical and practical insights for evaluating ecological resilience in similar ecologically vulnerable regions.The research findings offer a scientific foundation for ecological protection and restoration in the Wuling Mountain Area, and are crucial for advancing sustainable development in environmentally vulnerable regions.
Point 6. Conclusion: instead of repeating the most important results, I propose to summarise the new approach that has been used and to mention selected factors and their use.
Response 6:Thanks very much. we have revised the conclusions section in line with your comments and the section has been revised as follows.
This paper utilized NDVI as a proxy to assess ecological resilience. The ecological resilience index was calculated based on NDVI data spanning from 2000 to 2020. The study analyzed the spatial and temporal variability of ecological resilience in the Wuling Mountain Area and investigated the factors influencing ecological resilience using ge-odetector. The main conclusions are as follows: (1) The growth of ecological resilience in the Wuling Mountains subregion fluctuated slowly, with an overall upward trend. In addition to the Hubei subregion, the ecological resilience indices of the three subregions of Guizhou, Chongqing and Hunan also showed an overall upward trend.(2) The ecological resilience of the Wuling Mountain Area is dominated by medium and medium-high levels. The spatial distribution of the ecological resilience index in the Wuling Mountain Area and the three sub-areas of Guizhou, Hunan and Hubei varies considerably, while changes in the Chongqing sub-area are relatively small. (3) Differences exist in the elements that affect the ecological resilience of the Wuling Mountain Area and its four sub-areas. Climatic factors like as average yearly temperature and annual precipitation significantly impact the ecological resilience of the Wuling Mountain Area, and population density, GDP, the proportion of built-up land area and carbon emissions have a significant influence on the ecological resilience of the four sub-areas.
Suggestions for enhancing ecological resilience in the Wuling Mountain Area are provided based on the research findings on affecting factors.
(1)The Wuling Mountains region should strengthen the monitoring and predic tion of climate change and formulate climate adaptation and ecological restoration strategies.
(2)The Wuling Mountain Area should formulate scientific urban planning to optimise the population layout, strengthen the public's awareness of environmental protection and work together to maintain the ecological environment.
(3)The Wuling Mountain Area should adjust its economic structure to promote industrial transformation, develop a green economy and a low-carbon economy, and promote environmental sustainability in the region.
(4)When constructing construction land in the Wuling Mountain Area, it is important to prioritize the protection and restoration of the ecological environment to minimize harm.
(5)The Wuling Mountains region should promote technological innovation and clean energy to reduce carbon emissions.
We hope the above revisions can meet the requirements of this journal.
If there is anything else we should do, please do not hesitate to let us know.
Once again, we appreciate your valuable comments.
Best regards,
Shuiliang Liu
Round 2
Reviewer 1 Report
Comments and Suggestions for Authors
The authors' current revision is suitable for publication. However, the authors need to carefully revise the Materials and Methods, as well as the Results and Discussion Part one, which has a high degree of overlap with the existing literature.
Author Response
Dear Reviewer:
Thank you for your comments concerning our manuscript. These comments are all valuable and helpful for improving our article. We have carefully read your comments and have tried our best to revise the manuscript. And point-by-point responses to your comments are listed below.
Point 1. The authors need to carefully revise the Materials and Methods, as well as the Results and Discussion Part one, which has a high degree of overlap with the existing literature.
Response 1: Thanks for your suggestion, the Materials and Methods, as well as the Results and Discussion Part one has been revised in the text as requested.
We hope the above revisions can meet the requirements of this journal.
If there is anything else we should do, please do not hesitate to let us know.
Once again, we appreciate your valuable comments.
Best regards,
Shuiliang Liu

Reviewer 2 Report
Comments and Suggestions for Authors
In view of what the authors have exposed, I am recommending this study for publication.
Author Response
Dear Reviewer:
Thank you for your comments concerning our manuscript. These comments are all valuable and helpful for improving our article. we thank you for recognising this study.
Best regards,
Shuiliang Liu